

# A multi-instrument fuzzy logic boundary-layer top detection algorithm

Elizabeth N. Smith[1] and Jacob T. Carlin[2,1]

[1]NOAA/OAR National Severe Storms Laboratory, Norman, Oklahoma, USA

[2]Cooperative Institute for Severe and High-Impact Weather Research and Operations, The University of Oklahoma, Norman, Oklahoma, USA

**Correspondence:** Elizabeth N. Smith (elizabeth.smith@noaa.gov)

**Abstract.** Understanding the boundary-layer height and its dynamics is crucial for a wide array of applications spanning various fields. Accurate identification of the boundary-layer top contributes to improved air quality predictions, pollutant transport assessments, and enhanced numerical weather prediction through parameterization and assimilation techniques. Despite its significance, defining and observing the boundary-layer top remains challenging. Existing methods of estimating the boundary-

layer height encompass radiosonde-based methods, radar-based retrievals, and more. As emerging boundary-layer observation platforms emerge, it is useful to reevaluate the efficacy of existing boundary-layer top detection methods and explore new ones.

This study introduces a fuzzy-logic algorithm that leverages the synergy of multiple remote-sensing boundary-layer profiling instruments. By harnessing the distinct advantages of each sensing platform, the proposed method enables accurate boundary-layer height estimation both during daytime and nocturnal conditions. The algorithm is benchmarked against radiosonde-

derived boundary-layer top estimates obtained from balloon launches across diverse locations in Wisconsin, Oklahoma, and Louisiana during summer and fall. The findings reveal notable similarities between the results produced by the proposed fuzzy-logic algorithm and traditional radiosonde-based approaches. However, this study delves into the nuanced differences in their behavior, providing insightful analyses about the underlying causes of the observed discrepancies. The fuzzy-logic boundary-layer top detection algorithm, called BLISS-FL, is released publicly fostering collaboration and advancement within

the research community.

## 1 Introduction

The atmospheric or planetary boundary layer (BL) is often defined as the layer of the atmosphere directly influenced by Earth's surface. This is the portion of the atmosphere where the bulk of human socioeconomic activity takes place, yet it is also one of the least observed portions of the atmosphere. Given the depth of the atmosphere and common obfuscation by clouds (e.g.,

McGrath-Spangler and Denning, 2013), satellite-based observing of the BL remains difficult and uncertain. Commonly and consistently available observations include surface meteorological conditions and weather radar observations, which both may provide indirect inferences about BL conditions. The only widely available direct observations of the BL in the United States are radiosondes, which typically are only launched twice per day at 0000 and 1200 UTC (European nations, for example, have developed a diverse network of lower atmospheric sensors; see e.g., Cimini et al. 2020; Rüfenacht et al. 2021). Operational





radiosonde sites are spaced at best hundreds of kilometers apart (Melnikov et al., 2011), which is insufficient for representing conditions on meso- to BL-scales. This lack of observation in the BL has been coined a "data gap" (Bell et al., 2020) and several community reports and surveys have long called for improvement (National Research Council, 2009, 2010; National Academies of Sciences, Engineering, and Medicine, 2018a, b) to serve the diverse socioeconomic needs of modern society.

The BL evolves on a diurnal cycle, as shown in Fig. 1. As the sun rises (depicted as occurring near 1200 UTC in Fig. 1)
the earth's surface is warmed, which through heat (and moisture) fluxes leads to turbulence processes that mix the atmosphere above the surface. As these processes continue and grow, the well-mixed region above the surface becomes deeper developing the BL. The entrainment zone separates the BL from the free atmosphere and is characterized by transfer of mass and momentum across this boundary. The BL height continues to increase throughout the day as the convective mixed layer grows (barring disruption by other forcing such as an air mass change or strong advection) until the sun begins to set (depicted as occurring
near 0130 UTC in Fig. 1). The evening transition is characterized by the decay of the previous day's turbulent mixing. As mixing slowly shuts down and the surface loses heat to the atmosphere, the surface typically becomes cooler than the atmosphere above it, setting up a shallow surface-based inversion. This marks the beginning of the nocturnal BL or nocturnal surface layer. Above this layer, the residual layer retains characteristics of the previous day's boundary layer and is topped with the capping inversion (remnants of the entrainment zone). In conditions that allow for continued cooling, the nocturnal surface layer grows
until the sun rises again, repeating the process on another day.

Knowledge of the BL height and its evolution is central to many high-impact applications. These include air quality transport and dispersion forecasts (e.g., Dabberdt and Coauthors, 2004), fire weather monitoring and evolution (e.g., Clements and Coauthors, 2007), and the initiation of deep moist convection (e.g., Browning, 2007). Additionally, as numerical weather prediction models play an increasingly large role in the forecast process, real-world BL height observations are crucial for constraining
planetary BL parameterization schemes (e.g., Cohen et al., 2015), which can have a pronounced effect on subsequent convection forecasts (e.g., Crook, 1996; Stensrud and Weiss, 2002; Cohen et al., 2017) but have been shown to often exhibit large errors compared to observations (e.g., Grimsdell and Angevine, 1998). Recently, Tangborn et al. (2021) assimilated BL height observations and found positive impacts on temperature and winds in the simulated afternoon convective BL.

A variety of observation platforms are used to measure BL processes. For example, Doppler radar on various wavelengths
(Gal-Chen and Kropfli, 1984; Minda et al., 2010; Banghoff et al., 2018; Duncan et al., 2019), radar wind profilers (e.g., Ecklund et al., 1988; Rogers et al., 1993), microwave radiometers (e.g., Troitsky et al., 1993; Djalalova et al., 2022) and interferometers (e.g., Feltz et al., 1998; Knuteson and Coauthors, 2004; Turner and Löhnert, 2014), lidar-based instruments (e.g., Grund et al., 2001; Froidevaux et al., 2013; Spuler et al., 2021), spaceborne platforms (e.g., McGrath-Spangler and Denning, 2012), instrumented towers (e.g., Fernando and Coauthors, 2019), and both piloted (e.g., Hane et al., 1993) and
unpiloted (Segales et al., 2020) aircraft are all common tools used to understand the BL and its structure and processes. Myriad methods exist specifically for deducing BL height. Estimation based on data from balloon-borne radiosondes is one widely used method (Seidel et al., 2010). Seemingly straightforward, this method has drawbacks such as poor spatial resolution as discussed previously and discrepancies between the exact profile techniques used to find the BL height. Ceilometers are often integrated into Automated Surface Observing System (ASOS; NOAA et al., 1998) stations in addition to common deployment for research





purposes (e.g., Uzan et al., 2016). Aerosol backscatter as measured by ceilometers can be used in various techniques—some
      provided by instrument manufacturers, some developed by operators—to estimate BL height (Caicedo et al., 2017). Even
      within one manufacturer-provided method, BL height estimates can be ambiguous and require additional analysis, as Caicedo
      et al. (2017) found with the CL31 celiometer. Radar wind profilers can provide multiple products from a single platform,
      which have been combined in fuzzy logic algorithms to estimate BL height (Bianco and Wilczak, 2002, 2008). This approach

demonstrates how combining multiple measurements including those of processes indirectly related to BL height may be useful
      in an estimation technique. However, radar wind profilers typically do not have very high temporal resolution and are not as
      small in form factor as some other BL observation platforms such as lidars.

      Bonin et al. (2018) proposed a fuzzy logic algorithm for determining BL height[1] from Doppler lidar data and found promis-
      ing results using data from the Indianapolis Flux Experiment (INFLUX; Davis and Coauthors, 2017). This approach is ad-

vantageous in that it combines multiple estimates of BL height, provides a measure of uncertainty for each estimate, and is
      adaptable to users' individual needs and cases. However, no thermodynamic information is incorporated, making the approach
      most-suited for mechanically induced mixing rather than BL height development related to buoyancy processes such as noc-
      turnal stable boundary layers. Additionally, this method relies on a single instrument which can limit the independence of the
      multiple input variables to the algorithm.

We hypothesize that the capability and applicability of such a fuzzy logic approach to BL height estimation could be ex-
      panded by incorporating multiple instrument datastreams. In recent decades several integrated platforms and sites have been
      developed, which combine BL thermodynamic and kinematic profiling capabilities such as the such as the Department of
      Energy Atmospheric Radiation Measurement Southern Great Plains site (Sisterson et al., 2016) and multiple mobile facilities
      (e.g., Karan and Knupp, 2006; Knupp et al., 2009; Wingo and Knupp, 2015; Wagner et al., 2019). A team of scientists at the

University of Oklahoma (OU) and NOAA's National Severe Storms Laboratory (NSSL) developed and continue to operate
      the Collaborative Lower Atmospheric Mobile Profiling Systems (CLAMPS1 and CLAMPS2). Designed as sibling platforms
      OU-NSSL CLAMPS1 and NOAA-NSSL CLAMPS2 combine a Doppler lidar (DL), atmospheric emitted radiance interferom-
      eter (AERI), and microwave radiometer (MWR) into a single facility (see Table 1) capable of profiling temperature, moisture,
      horizontal wind, and vertical velocity with vertical resolution on the order of tens of meters and temporal resolution on the

order of seconds to minutes (depending on the quantity of interest). Such integrated platforms provide a unique opportunity to
      explore multi-instrument value-added products. In the case of CLAMPS, these value-added products have the benefit of high
      temporal resolution, which can be critical for some applications. Recognizing the variability and strengths and weaknesses
      of various remote sensing BL height detection methods, we propose to use the instruments onboard CLAMPS together in a
      fuzzy logic approach for BL height estimation including thermodynamic BL profiles, which should extend this approach to

the overnight hours in at least some conditions. Herein, we expand upon the work of Bonin et al. (2018) and introduce a new
      multi-instrument BL height detection algorithm.

---

[1]In Bonin et al. (2018), the term mixing-layer height or mixing height is used, not BL height. Their analysis and algorithm is limited to only lidar data, and
thus is only evaluating mechanical mixing processes in the boundary layer. To avoid confusion, we use BL height expecting that this definition is inclusive of
mixing processes.



## 2 Fuzzy logic algorithm

Fuzzy logic methods are used to discern and classify signals based on some known characteristics (Mendel, 1995). Fuzzy logic is unlike Boolean logic in that is allows for degrees of "truth" opposed to the binary "true" or "false." The fuzzy logic process

relates input variables to an output characteristic via so-called membership functions, which vary from zero (which indicates the input variable is not a member of a given class) to one (which indicates the input variable is a member of a given class). By taking a weighted mean, the membership values of all input variables are aggregated and then defuzzified to determine the desired characteristic of the measurement.

Fuzzy logic methods have been applied in many ways within atmospheric science before the work of Bonin et al. (2018).

These approaches have been popular amongst radar scientists and developers, where it has been applied for identification of non-precipitation echos and radar artifacts (Gourley et al., 2007; Mahale et al., 2014), hydrometeor classification methods (Vivekanandan et al., 1999; Liu and Chandrasekar, 2000; Park et al., 2009), and detection of precipitation modes (Yang et al., 2013). Applying the method to Doppler lidar measurements was a novel component of the Bonin et al. (2018) work. Kotthaus et al. (2023) pointed out the potential synergy of combining multiple methods or observations from multiple sensors. Our

proposed algorithm uses fuzzy logic to combine high resolution boundary layer profiler observations from multiple instruments toward that aim.

We now describe and demonstrate our newly developed algorithm using CLAMPS1 observations collected in Norman, Oklahoma on 25 June 2020, shown in Fig. 2. Much of the initial framework closely follows that of Bonin et al. (2018, hereafter B18), with any deviations from or expansions upon B18 highlighted. The fuzzy logic algorithm takes a two-step approach: a

first-generation BL height estimate and a second-generation BL height estimate. In the first-generation step, only measurements of BL mixing processes (i.e., measurements that show mixing is ongoing) are included. Given the instruments available on CLAMPS, these measurements include turbulence information from the DL. The second-generation estimate additionally includes indicators of mixing processes (i.e., measurements that show that mixing has occurred). For example, a well-mixed potential temperature profile from the CLAMPS thermodynamic profilers would indicate BL mixing has occurred.

In the first-generation step, vertical velocity variance, $\overline{w'^2}$ (Fig. 2a, a1), as computed from the DL vertical stare observations, and the high-frequency vertical velocity variance, $\overline{w'^2_{HF}}$ (Fig. 2b), are included as the indicator variables. The method for computing $\overline{w'^2_{HF}}$ is detailed in B18; we apply it here to control for any potential contribution to vertical velocity variance signals due to non-turbulent motions such as waves, drainage flows, and sub-meso motions (Bonin et al., 2017). B18 includes several other variables from the more complex Doppler lidar scan patterns available in their dataset; we include only $\overline{w'^2}$ and

$\overline{w'^2_{HF}}$ as indicators. The membership functions are defined following B18 as half-trapezoids, where parameters $x_1$ and $x_2$ are the values at which the function increases above zero and reaches a maximum of one, respectively. For $\overline{w'^2}$, $x_1 = 0.02$ and $x_2 = 0.08 \text{ m}^2 \text{ s}^{-2}$. For $\overline{w'^2_{HF}}$, $x_1 = 0.0025$ and $x_2 = 0.01 \text{ m}^2 \text{ s}^{-2}$. In the fuzzification step, data are re-gridded via interpolation onto a common 10-m by 10-min grid and evaluated via the membership function, becoming a standard shaped field of values from zero to one reflecting degree of membership in the BL. Both input variables are given equal weights of one.





At this stage the algorithm deviates from B18. Since the first-generation estimate depends only on measures of mixing, it relies only on mechanically induced turbulence and mixing to determine BL height. Buoyant processes also play a role in BL development, and are often a dominant process at night when stable boundary layers are more common. In later steps the BL height estimate from the first-generation step is used as a type of constraint on the second-generation step. Thus if there is a complete failure to detect a buoyancy-driven BL height in the first-generation step, the second-generation step is unable to

recover. In order to capitalize on the availability of thermodynamic profiles and extend the capability of our algorithm into the overnight hours, the first-generation step also includes temperature inversion height as an input variable (Fig 2c). Note this is not a *measure* of mixing. To limit the effect of this additional input beyond periods where buoyancy is more likely to dominate mechanical generation of mixing (e.g., nocturnal stable periods), a time-dependent weighting function is defined based on the local sunrise and sunset time (Fig 2d). This weighting function allows the inversion height to have an effect

on the algorithm during the overnight hours with sloped increasing and decreasing weights during the evening and morning transitions, respectively. The membership function in this case is step wise, and thus not authentically a fuzzified field. All levels above and below the inversion height are assigned membership values of zero and one, respectively.

Now that all the first-generation variables are in place, they are aggregated together by taking a weighted mean. Recall that $\overline{w'^2}$ and $\overline{w'^2_{HF}}$ both are assigned a weight of one. The inversion height has a time-variable weight ranging from zero during the

day to two during the night. The result is one aggregate field representing the degree of membership in the BL based on these input variables. For our example case, the first-generation aggregate is shown in the upper panel of Fig. 3.

The top of the boundary layer is defined as the first level where the aggregate value is less than or equal to 0.5, following B18. Using only the first layer where this condition is met guarantees the identified level is connected to the surface and not an elevated layer. Using the above described definitions and constraints, the first-generation estimates of the boundary-layer top

are compiled. These values will be used in the second-generation step as constraints within the fuzzy membership functions.

In the second-generation step, profiles of signal-to-noise (SNR) ratio, SNR variance, $u$-component wind, and $v$-component wind from the Doppler lidar are all used, consistent with B18. This algorithm also includes information from CLAMPS thermodynamic retrievals in the form of potential temperature and water vapor profiles. For all of these profiles, this algorithm dynamically defines the membership function for each profile (each time an observation is available) using the same approach

as B18; all profiles are passed through a discrete Haar (Haar, 1910) wavelet transform to detect gradients and membership is defined based on those gradients. The use of Haar wavelets to identify gradients in lidar backscatter is not new (e.g., Cohn and Angevine, 2000; Davis et al., 2000; Brooks, 2003); we extend the approach to thermodynamic profiles here.

For each profile considered, a discrete Haar wavelet transform is applied to the profile at each time. The sensitivity of the Haar wavelet dilation was examined and determined to be low in this application; a default value of 100 m was used here. In

the case of $u$- and $v$-component profiles, the transform is applied to each profile separately, then the the vector magnitude of the wavelet transform is used in the membership function computation process to follow, following B18. In all cases, after the transform is applied the five peaks of greatest magnitude are identified. The height of each of those peaks is evaluated against the height of the first-generation boundary-layer top estimate. Only peaks within 25% of that estimate are retained (i.e., the



aforementioned constraint). The membership function is then defined based on the remaining peaks:

$$
\quad M(z) = \begin{cases} 1, & \text{if } z \leq z(P_{min}) \\ (1,0); \text{ decreasing by } \frac{P}{sum(P_{all})}, & z \geq z(P_{min}) \text{ and } z < z(P_{max}) \\ 0, & \text{if } z > z(P_{max}) \end{cases} \tag{1}
$$

where $M$ is the membership function, $P$ is an identified and retained peak, and $z$ is height above the surface. If no peaks are identified, then no membership function is generated for the given input and it is not included in the aggregate at that time. This method of defining the membership function is considered dynamic since it is repeated for each unique profile, allowing it to change as a function of time.

The second-generation aggregate is produced like the first—by employing a weighted mean. As mentioned previously, the membership values from the first-generation step are included in this mean with a weight of one (except the inversion height; the time-varying weight still applies). Second generation variables are all given a weight of two. Again, the first level where the aggregate value is less than or equal to 0.5 is used as the threshold to define the boundary-layer top. The complete fuzzy logic-based BL top detection algorithm is summarized in flowchart form in Fig. 4. The second-generation estimate of boundary-layer

top is the final estimate and is shown for the example case in the bottom panel of Fig. 3.

     Since all data were fuzzified onto a common 10-min by 10-m grid, fuzzy logic boundary-layer top estimates are provided every 10 minutes. Given that we know the BL grows and decays as a physical process, we can use information about the points surrounding a given point to both smooth extreme variability in the estimate (which could arise as artifacts of observation or algorithm limitations) and provide a measure of that variability for users. A centered triangle window method (with a width of

one hour) is applied to the 10-min estimates of BL height. Using triangle weighting, a weighted mean is used to smooth the BL height estimates in time. To preserve information about variability, all samples within the centered window are included to compute standard deviation representative of that mean. The centered triangle window slides from sample to sample, retaining the 10-min resolution of the provided dataset. The provided output also provides data on a centered hourly time axis for a more simple comparison to standard observation platforms such as radar wind profilers, ceilometers, and radiosondes, which are

more often available precisely on the hour.

## 3   Data

To understand the potential value of our algorithm combining high-resolution boundary-layer profiler observations from multiple instruments, it is critical that we examine the algorithm's output in comparison to reference data. Radiosonde observations are a commonly available and known source of data that can be used to estimate BL height, though not without uncertainty

(Seidel et al., 2010; Kotthaus et al., 2023). This uncertainty is explored in order to provide a reference dataset for comparison to the proposed fuzzy logic algorithm for BL height. Three observation periods during which CLAMPS observations and radiosonde data are available concurrently are used.



## 3.1 CHEESEHEAD (2019)

The Chequamegon Heterogeneous Ecosystem Energy-balance Study Enabled by a High-density Extensive Array of Detectors (CHEESEHEAD) experiment was conducted near Falls Park, Wisconsin in mid-summer to early fall 2019 (Butterworth et al., 2021). Primarily sponsored by the National Science Foundation, additional support from NOAA enabled the concurrent deployment of CLAMPS1 and CLAMPS2 at Lakeland and Prentice Airports, respectively, for approximately five weeks of data collection. CHEESEHEAD also included radiosonde launches near the WLEF-TV 400-m very tall tower site, which was approximately 45 km away from both CLAMPS locations (north of Prentice, west of Lakeland). The analysis presented in this work will focus on CHEESEHEAD observations collected from 19 September 2019 to 10 October 2019.

The CLAMPS1 (located at Lakeland Airport, 45.92° N, 89.73° W) DL collected plan position indicator (PPI) scans at 70 ° elevation every 20 minutes, and remained in a zenith pointing mode otherwise. The DL provides range-resolved, line-of-sight measurements of radial velocity, intensity (signal-to-noise ratio [SNR]+1), and attenuated backscatter. In the case of PPI scans meant for velocity azimuthal display (VAD) analysis, these data are post-processed to produce profiles of horizontal wind speed and direction. The zenith pointing scans are used for vertical velocity information, and are post-processed to provide turbulence information such as vertical velocity variance. The DL was configured to provide profiles with 18-m vertical spacing. The CLAMPS1 MWR and AERI were both deployed, and thermodynamic profiles as retrieved by the TROPoe physically based algorithm (previous versions known as AERIoe; Turner and Löhnert, 2014) were made available every ten minutes. For different user needs and use cases we provided retrievals using only AERI data, only MWR data, and combining the data from both the AERI and the MWR in the TROPoe retrieval algorithm. In this study we use the combined AERI+MWR thermodynamic retrieval.

The CLAMPS2 facility (located at Prentice Airport, 45.54° N, 90.28° W) operated similarly, but the DL collected PPIs at 60 ° elevation every 5 minutes. The onboard MWR was deployed similarly, but CLAMPS2's AERI was damaged in transit and did not operate for CHEESEHEAD. As such, CLAMPS2 retrievals are limited to MWR-only. For the presented analysis herein, we chose to exclude CLAMPS2 CHEESEHEAD data from this analysis in order to remove any potential differences resulting from data sources, as it would have otherwise been the only non-AERI based thermodynamic data source.

## 3.2 NWC/RIL (2020)

By summer 2020, planned CLAMPS field missions were cancelled due to the global SARS-COVID19 pandemic. Under the uncertainty of when those missions may resume or be rescheduled, a local deployment was organized between the the National Weather Center (NWC) and Radar Innovation Lab (RIL) properties in Norman, OK to evaluate instrument readiness (near 35.1816 ° N, 97.4398 ° W) This deployment included the CLAMPS1 platform, which operated from 4 June 2020 – 8 August 2020. This deployment location is within a short walk of the Norman, Oklahoma National Weather Service upper air release point. The DL conducted 70 ° elevation PPI scans providing horizontal wind profiles every 5 minutes and operated in vertical stare mode at all other times. The thermodynamic retrieval algorithm included both AERI and MWR observations; profiles are available every 10 minutes.



### 3.3 PBLTops (2020)

As part of an experiment seeking to develop this method and validate the dual-polarization radar-based boundary-layer detection method proposed in Banghoff et al. (2018), both CLAMPS platforms were deployed and continuously collected data from 21 August 2020 until 24 September 2020. During this period, CLAMPS2 remained stationary at the University of Oklahoma's North Campus (35.237° N, 97.463° W), 6.7 km north-northwest of the Norman, Oklahoma National Weather Service (OUN) upper air release point. In contrast, CLAMPS1 was deployed at the Kessler Atmospheric and Ecological Field Station (KAEFS; 34.984° N, 97.516° W), 29.6 km south-southwest of the CLAMPS2 site, from 21 August 2020 until 3 September 2020; for the remainder of the period, CLAMPS1 was deployed at the Shreveport National Weather Service Office in Shreveport, Louisiana (SHV; 32.452°N, 93.842° W) with collocated radiosonde launches. For both CLAMPS, the DL conducted 70 ° elevation PPI scans providing horizontal wind profiles every 10 minutes and operated in vertical stare mode at all other times. The thermodynamic retrieval algorithm included both AERI and MWR observations with profiles available every 10 minutes.

### 3.4 Radiosonde observations

High-resolution radiosonde observations were available for the CHEESEHEAD campaign (Vaisala research grade radiosondes processed by the National Center for Atmospheric Research; NCAR/EOL In-Situ Sensing Facility and University of Wisconsin-Space Science and Engineering Center (SSEC) 2019) and at OUN for the NWC/RIL and PBLTops campaigns (high resolution operational radiosonde data provided by National Weather Service, Norman). These soundings have a mean vertical resolution of approximately 5 m in the lowest 3 km and the exact launch time recorded. To prevent erroneous BL height estimates due to minor localized fluctuations in the vertical profiles (particularly for gradient-based calculations), the soundings were interpolated to a fixed grid with a 20-m vertical spacing and smoothed using a third-order Savitzky–Golay filter (Savitzky and Golay, 1964). Using all the CHEESEHEAD radiosondes—since they have constant vertical spacing already—all BL height estimation methods (discussed below) and the median of the estimates were compared on the original radiosonde vertical grid and with 100-m, 300-m, and 500-m smoothing windows applied (Fig 5). Overall, BL height estimates all collapse to the diagonal of the scatterplot, indicating one-to-one agreement of most datapoints. There is some scatter where smoothing allows for a deeper BL height estimate, but there are no clear clusters or patterns between 100-m, 300-m, and 500-m smoothing windows. Note that the 100-m window points are mostly obscured by overlapping with other points—they are only noticeable by the careful comparison of shades of other markers. Since these comparisons of smoothed BL height estimates showed weak-to-no sensitivity to smoothing-window size, the data in this study are smoothed using a 300-m window.

At SHV during PBLTops and whenever high-resolution data were otherwise unavailable, coarse publicly available radiosonde observations were used instead. No smoothing was performed for the coarse soundings. Because exact launch times were not available for the coarse soundings, a launch time 1-h before the nominal time was assumed as per National Weather Service manual documentation of observation procedures (e.g., soundings denoted as 1200 UTC were assigned a launch time of 1100 UTC; National Weather Service 2010).





Radiosondes were used to estimate BL height as a comparison for the fuzzy-logic algorithm output. Numerous methods for estimating the BL height from radiosonde data have been proposed. BL height is a parameter which tends to be defined based on the application and data availability at hand. A one-size-fits-all definition has not been agreed upon in the meteorological community, its sub-communities, or tangential communities. Here, we utilize an ensemble of seven BL detection methods following Seidel et al. (2010). Our reasoning here is two-fold. On one hand, it is useful to know if the fuzzy logic algorithm tends to match some methods more than others. This may mean the algorithm is more applicable in some settings or some important information is missing. On the other hand, and perhaps most interestingly, having an ensemble of estimates of BL height estimates provides a pseudo-measure of spread or uncertainty in the radiosonde estimation process itself. Large disparities between the given methods may suggest a more complex BL structure, increasing the level of difficulty for automated BL height detection.

The first of the seven radiosonde-based methods described in Seidel et al. (2010), also known as the "parcel method", finds the height at which the profile of virtual potential temperature $\theta_v$ becomes equal to the surface value. The second method extends this by adding 0.6 K to the surface $\theta_v$ value to prevent erroneously shallow BL estimates during the evening transition (Coniglio et al., 2013). The third, fourth, and fifth methods find the height of the maximum gradient of potential temperature $\theta$, minimum gradient of water vapor mixing ratio $q_v$, and minimum gradient of relative humidity, respectively. The last two methods are inversion-based and find the top of any surface-based temperature inversion and the bottom of the lowest elevated temperature inversion layer, respectively. These methods are summarized in Table 2. To achieve an overall singular estimate of BL height from each sounding, the median of all available estimates is taken. To understand variability across the methods, the 25th and 75th percentile BL height estimate values are retained from the suite of methods. This approach was chosen as an outlier-resistant way to gather information about the variability among radiosonde-based BL height estimation methods.

Seidel et al. (2010) found that while coarse data can be sufficient to detect BL height, the use of high-resolution data can change the estimate in statistically significant ways. A comparison of the medians of radiosonde-derived BL heights for both high-resolution and coarse data is shown in Figure 6. This comparison appears to support the Seidel et al. (2010) findings. While many of the afternoon and early evening soundings are close to the one-to-one line, several of the morning soundings fall to the right and below the one-to-one line, suggesting the high-resolution based estimates are lower than the coarse estimates. The 'error' bars, which represent the spread (i.e., interquartile range) of BL heights detected by the seven methods, also appear to cover a wider range for the high-resolution soundings. Further examining the comparison by looking at individual methods separately, we find slightly different results than those in Seidel et al. (2010). Those results showed methods that were surface-up computed such as parcel- and inversion- based methods were most sensitive to profile resolution-based errors, while Fig. 7 suggests that gradient-based methods are most sensitive in our dataset. Analyzing the methods separately highlights that using high-resolution sounding data tends to lead to lower median BL height, but the ranges and interquartile range values do not change much between coarse and high resolution datasets. There is no indication from this analysis that using high resolution data necessarily yields more accurate BL height values, and often users have access only to data that we have classified here as coarse resolution (i.e., publicly available and accessible National Weather Service radiosonde datasets). Our intention with this analysis is to understand and consider possible implications of including different resolution sounding datasets. Moving



forward, we use high resolution sounding data when it is available. If it is not available a coarse resolution radiosonde profile is used instead.

## 4   BL height estimate comparisons

Now we aim to understand how the fuzzy logic algorithm applied to CLAMPS data performs by comparing the resulting BL height estimates to BL height estimates from available radiosonde data. We use all the periods described in Sect. 3, always using high-resolution radiosonde profiles unless they are not available (e.g., some OUN launches during NWC/RIL and PBLTops; all SHV launches). All instances when a radiosonde and fuzzy logic BL estimate are available at the same time (within $\pm$ 30 minutes) and place are shown together in Fig. 8. From this bulk approach, a few things become immediately apparent. The fuzzy logic method applied to CLAMPS observations is more likely to estimate a lower BL height than radiosondes. This is especially true in the afternoon hours, which correspond to 1800 and 2100 UTC for the areas where these observations were collected. No 2100 UTC BL height comparisons (shown in yellow in Fig. 8) fall on or below the one-to-one line. The majority of 1800 UTC and most 0000 UTC BL height comparisons also trend above the one-to-one line, suggesting the fuzzy logic method applied to CLAMPS observations estimates a shallower BL. Earlier hours (0600 and 1200UTC) do not demonstrate as much of a signal.

In Sect. 3.4, examination of the methods summarized in Seidel et al. (2010) highlighted potential variability in BL height estimation based on radiosondes. Fig. 8 shows a comparison of the fuzzy logic BL height estimates to the median estimates from all radiosonde-based methods. In Fig. 9, the same comparison is repeated for each method independently. Root mean square error (RMSE), correlation ($r$), and bias (computed for all values and computed only for instances when the BL height is 1 km or greater) are also shown. Upon visual inspection of the different methods, the comparisons between BL heights from radiosondes and BL height estimates from the fuzzy logic algorithm applied to CLAMPS observations take on different characteristics. For example, the modified parcel method (Fig. 9b) shows most BL height comparisons (not occurring at 1200 UTC) collapsing toward the one-to-one line, indicating agreement between the approaches about the BL height. The correlation value when comparing fuzzy logic BL height to radiosonde-based BL height using the modified parcel method is the highest of all methods at 0.835. Conversely, using the elevated inversion method (Fig. 9g) in the comparison results in radiosonde-based BL height estimates nearly always being higher than those derived from the fuzzy logic algorithm applied to CLAMPS observations. In this instance, the correlation value is the lowest of all methods at 0.338. Selecting any one of these methods for deriving BL height from radiosondes could impact our understanding of the performance of the fuzzy logic algorithm applied to CLAMPS, since it would define the baseline against which the algorithm is compared. No one method is necessarily known to be more accurate than the others. In the absence of a well-justified or well-known definition of the top of the boundary layer that can be applied here, we choose to move forward with the median of all methods, as shown in Fig 8. We retain information about the range of BL height estimates produced by all the methods in the form of the interquartile range of BL height values. This approach allows us to have an outlier resistant way to measure large variability across the radiosonde-based methods, which can suggest uncertainty in the radiosonde-based BL height estimate.



In Fig 8, every available radiosonde–CLAMPS BL height estimate pair is compared without consideration of the atmospheric conditions at the time of the observation. In order to better understand the potential causes of discrepancies between the radiosonde and CLAMPS-based BL height estimates and ensure a robust comparison, each matched pair was manually interrogated for inclusion or exclusion in a similar approach as Banghoff et al. (2018). In our case, criteria for exclusion were

developed to describe instances in which discrepancies between CLAMPS and radiosonde BL heights may not necessarily reflect the intrinsic performance of the proposed CLAMPS algorithm. These criteria were (1) ambiguous cases where the BL height was unable to be confidently determined from the radiosonde data; (2) cases where CLAMPS observations were not able to be collected over a deep enough layer to capture the likely full depth of the BL (primarily DL observations due to a lack of scatterers); (3) cases where the BL top (e.g., entrainment layer or capping inversion) was deep and the radiosonde and

CLAMPS methods identified different parts of this transition region; and (4) cases with complex/non-canonical BL structures. To identify cases for exclusion, both authors independently evaluated each time-matched CLAMPS and radiosonde profile to determine whether one or more of these criteria were met without respect to how their BL height estimates compared, then discussed and reconciled any differences. These criteria were used to successively exclude data pairs and get a better sense of algorithm performance. As each criteria was applied successively, more pairs were removed from the comparison dataset as

summarized in Table 3. Starting with 168 data pairs, 104 remained after all criteria were applied. The same set of statistics that were examined in the method comparison ($r$, bias, RMSE) are computed for each successively reduced dataset and shown in Table 3.

Figure 10 shows the comparison with sequential exclusions of criterion (1) (Fig. 10a) and criteria (1) and (2) (Fig. 10b). Removing the cases where the BL height was ambiguous in the radiosonde data, criterion (1), does not make much visual

impact on the comparison. The pairs that are excluded follow no obvious pattern. They likely should not be expected to; cases where defining the BL height was ambiguous should lead to the CLAMPS-based method both over- and under- estimating BL height compared to the radiosonde estimates without any physically defined trend. The same is not true for cases where the BL is too deep for the CLAMPS platforms to collect measurements over its full depth (criterion (2), (Fig. 10b)). These pairs are primarily found on the upper left side of the comparison diagram, meaning CLAMPS-based methods are underestimating these

pairs. These cases are occurring in conditions where the CLAMPS observing systems simply cannot observe to a high-enough level above the surface. The algorithm's first step relies most heavily on DL-observed variables. If the BL is quite deep, or for a variety of reasons (e.g., recent precipitation or other airmass characteristics) the scatterer load is low, the DL will not be able to reach levels above the surface where the BL height may reside. It is also true that the effective resolution of the thermodynamic profiles available from the platforms onboard CLAMPS decay with increasing height above the surface. An

exceptionally deep BL is simply difficult for the instruments onboard to sample, and therefore the algorithm will also not be able to retrieve those BL height values well. With these pairs removed the comparison overall improves. Moving to a more quantitative comparison made available to us via statistics computed in Table 3, we see that the statistics generally improve when criteria (1) is applied. The $r$ value modestly increases while bias and $RMSE$ both improve by 32.4 m and 85.9 m, respectively. When criteria (2) is applied the improvements are more apparent. The $r$ value increases further to 0.846. The

bias and $RMSE$ value improvements are the largest when this criteria is applied compared to any of the other criteria; bias



improves by 80.7 m and $RMSE$ by 108.6 m. The pairs removed by criteria (2) are concentrated at greater BL height values likely resulting in large improvements to statistics like bias. On the other hand, removals based on criteria (1) did not appear to follow any pattern, so improvements were distributed resulting in more subtle impacts to the statistics.

Figure 11 shows the comparison with sequential exclusions of criteria (1)-(3) (Fig. 11a) and all criteria (Fig. 11b). Compared
to Fig. 10b, Fig. 11a, which also excludes cases where the radiosonde and CLAMPS-based methods identified different parts of the BL-top, shows fewer pairs in the area just above the one-to-one line. This suggests that when the two methods detect different parts of the BL-top, the CLAMPS-based method is more likely to find a BL height closer to the surface. This makes sense as the fuzzy logic algorithm is a bottom-up algorithm that searches for the first point above the surface where the membership value crosses a threshold. As discussed previously, there is variability among chosen methods in how BL height
is estimated from radiosonde data, and this is a scenario in which that variability can be important. Finally, Fig. 11b shows the comparison with all exclusion criteria applied (now additionally excluding complex BL structures). Compared to Fig. 11a, there are not obvious differences. The additional exclusions occur both above and below the one-to-one line. As for the ambiguous BL cases, this makes some sense as complex BL structures should not be expected to follow specific patterns and we see the CLAMPS-based method both over- and under- predict BL height for these cases.

With all the exclusions applied, we can use the final set of remaining pairs shown in 11b as the so-called best cases for comparing the fuzzy logic algorithm to radiosonde-based BL heights. Differences in this comparison can be more directly attributed to characteristics of the proposed CLAMPS algorithm, or in other words these cases are those where the BL conditions are well-enough understood and represented by both sets of observation platforms for the comparison to be made. Many of the comparison pairs fall close to the one-to-one line, suggesting fairly good agreement between the fuzzy logic and radiosonde-
based BL heights. When the pairs are not close to the one-one-line they tend to be above it, which indicates that in some cases the fuzzy logic algorithm applied to CLAMPS observations still leads to underestimation compared to radiosonde-based BL heights. In the best case comparison, we still see that the overnight and morning pairs (i.e., 06 and 12 UTC) are clustered together in the lower left side of Fig. 11b, which makes it difficult to examine patterns more closely. To better understand how the fuzzy logic algorithm compares to radiosonde based BL height estimates, we broke the best case pairs with all ex-
clusions applied into overnight/morning pairs and afternoon/evening pairs (pairs occurring between 18 and 00 UTC). Again, $r$, bias, RMSE are computed, but now for these subgrouped pairs based on time of day (values are reported in Table 3). The overnight/morning group (of the best case pairs) is a larger group than the afternoon/evening group. The $r$ value is lower in the overnight/morning group while both bias and RMSE appear to be improved compared to the afternoon/evening group. This is misleading however, as a bias of 100 meters in the overnight or morning hours is much more impactful than during the
afternoon hours when the BL is fully developed and much deeper. In any case, these subgroups show different results than the full best case pairs group, suggesting that the time of day has an impact on the comparison between the radiosonde-based estimates and the fuzzy logic algorithm. With more frequent and regularly available radiosondes, this analysis could lead to deeper understanding about the impact time of day has on the performance of the fuzzy logic algorithm itself.



## 5 Summary and Outlook

In this work, we present a fuzzy logic approach for estimating BL height that incorporates kinematic and thermodynamic observation datastreams into a multi-instrument value-added product. This algorithm expands upon the work of Bonin et al. (2018), specifically by including thermodynamic profiles. The fuzzy logic algorithm follows a two-step process to produce BL height estimates using various input observations from the kinematic and thermodynamic observing platforms onboard the CLAMPS facilities. Output is produced every ten minutes along with standard deviations from an hour-wide centered

sliding window. While this algorithm was developed for use with the CLAMPS platforms, it could be applied to or adapted for similarly instrumented facilities such as but not limited to the Department of Energy's Atmospheric Radiation Measurement profiling facilities.

To characterize and understand how the presented algorithm performs, it is compared with radiosonde data collected from three periods: the CHEESEHEAD project in Wisconsin during the fall of 2019, the NWC/RIL deployment in Oklahoma

during summer 2020, and the PBLTops project in Oklahoma and Louisiana during summer and fall 2020. To make sure these comparisons are well suited, various impacts of radiosonde data resolution are examined (i.e., impacts of vertical spacing on gradient calculations and high-resolution versus coarse-resolution comparisons). This analysis does not aim to present the most correct method for computing BL height from radiosondes, but it shows the differences between methods and possible variability that can be introduced when including sounding datasets with different resolutions.

Comparisons are conducted in two ways, the first of which is a bulk comparison of all instances when a radiosonde-based estimate and fuzzy logic estimate of BL height are available at the same time (within $\pm$ 30 minutes) and place. Through this analysis we show that the fuzzy logic algorithm often results in lower BL height estimates than radiosonde-based methods (especially in the afternoon hours). When examining this comparison more closely, we find sensitivity to the exact BL height estimation method applied to radiosonde data as expected. As was the case for the analysis of radiosonde data resolution,

we do not intend to offer any one radiosonde-based BL height estimation method as preferred. In our application, using the median provides a basis for comparison and the 25th and 75th percentile BL height estimate values are retained from the suite of methods as an outlier-resistant measure of variability. However, this sensitivity to BL height estimation method (and potentially to resolution of radiosonde data itself, as noted above) presents a non-trivial complicating factor for any use case or comparing across past studies.

Comparisons are again conducted, but this time consideration is given to the atmospheric conditions at the time of the matched radiosonde-CLAMPS profile observation with the intention of understanding and eliminating extrinsic causes for discrepancies between the radiosonde-based BL estimates and results from the fuzzy logic algorithm. Each radiosonde-CLAMPS observation pair was manually interrogated and either included or excluded. Criteria for exclusion can be summarized as (1) ambiguous; (2) BL too deep for CLAMPS observation capability; (3) deep BL top (e.g., deep capping inversion or entrain-

ment layer); and (4) complex/non-canonical. The exclusions are applied successively, eventually leading to a final subset of best cases in which the BL conditions are well-enough understood and represented by both sets of observation platforms for a robust comparison to be made between the fuzzy logic algorithm and radiosonde-based BL height estimates. This compar-





ison suggests fairly strong agreement between the techniques with some cases in which the fuzzy logic algorithm applied to CLAMPS observations leads to an underestimation of BL height. Specific analysis is focused on the early morning and late

overnight periods. While some statistics included suggest perhaps even minor improvement compared to the daytime group, this may a misleading result. For example, a bias of 100 meters is more meaningful when BL height is on the order of 100 m, which is common in the overnight and morning hours compared to larger BL height values in fully developed daytime and afternoon BL. More data is needed in this comparison to understand the role time of day plays in how the fuzzy logic algorithm behaves. Unlike similar algorithms, it has the capability to utilize thermodynamic observation information and kinematic

observation information (when available) to provide BL height estimates throughout the diurnal cycle.

There are various techniques for observing the lower atmosphere, which individually offer an incomplete picture of the processes that characterize the BL. Methods of BL characterization, including BL height, can be subjective. These sources of ambiguity in understanding the BL can be addressed through synergistic combination of observation techniques. Kotthaus et al. (2023) describe the potential for instrument synergy to advance methodologies and products and to provide value in pro-

cess studies and applications. Specifically, they identify the potential of using multiple observing platforms to extend detection capabilities beyond those of individual platforms, suggested assessment of uncertainty among techniques, and employment of advanced retrieval techniques (e.g., fuzzy logic). The fuzzy logic algorithm presented in this work combines Doppler lidar, infrared spectrometer, and microwave radiometer observations to produce relatively high-resolution BL height estimates with an assessment of uncertainty or variability. While in its first testing iteration, the method has already been deployed in

research comparing BL height methods across research-grade tools during the CHEESEHEAD project (Duncan et al., 2022). It is currently in use for ongoing research related to several recent field programs and deployments on which CLAMPS has been deployed such as the TRacking Aerosol Convection interactions ExpeRiment (TRACER) and the American WAKE experimeNt (AWAKEN). The algorithm has been de facto integrated into the suite of tools frequent CLAMPS users implement in their own research, and there are plans to explore implementing it as part of the data system work flow as a value-added

product for standard CLAMPS operations. To facilitate further applications and use cases the fuzzy logic BL height detection algorithm implementation has been made available on a dedicated GitHub repository, BLISS-FL (Boundary-Layer height Inferred through multi Sensor Synergy-Fuzzy Logic; see code and data availability statement for access information). In the spirit of open science, making the algorithm available encourages collaboration, adaptation, and extension for ongoing algorithm refinement and enhancement. We invite readers to engage with the algorithm, contribute to its development, and apply it

to diverse problems and datasets.

*Code and data availability.* Observations used in this work are available on the CLAMPS archive hosted on the NSSL THREDDS server at https://data.nssl.noaa.gov/thredds/catalog/FRDD/CLAMPS.html. The fuzzy logic algorithm project is hosted in a GitHub repository at https://github.com/OAR-atmospheric-observations/bliss-fl.



*Author contributions.* ENS and JTC worked as collaborators throughout the process of this project from conceptualization, to data collection
and gathering, through algorithm development, testing, and refinement, to final analysis and writing. ENS conducted most CLAMPS data
analysis. JTC led development of statistics and visualization standards. ENS typically led first drafting of this article, but further writing,
reviewing, and editing were completed in collaborative partnership through frequent, at times weekly, working meetings.

*Competing interests.* The authors declare that they have no conflict of interest.

*Acknowledgements.* The authors acknowledge insights provided by Dr. Timothy Bonin of MIT Lincoln Labs, Dr. Tyler Bell and Dr. Josh
Gebauer of OU-Cooperative Institute for Severe and High-Impact Weather Research and Operations (CIWRO), and Katie Giannakopolous
of the University of Oklahoma. This work was prepared by the authors with support from the NSSL Forecast Research and Development
Division (ENS) and the NOAA/Office of Oceanic and Atmospheric Research under NOAA–University of Oklahoma Cooperative Agreement
NA21OAR4320204, U.S. Department of Commerce (JTC). The contents of this paper do not necessarily reflect the views or official position
of any organization of the U.S. Government. Portions of this work were supported by CIWRO Director's Discretionary Research Funds.



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





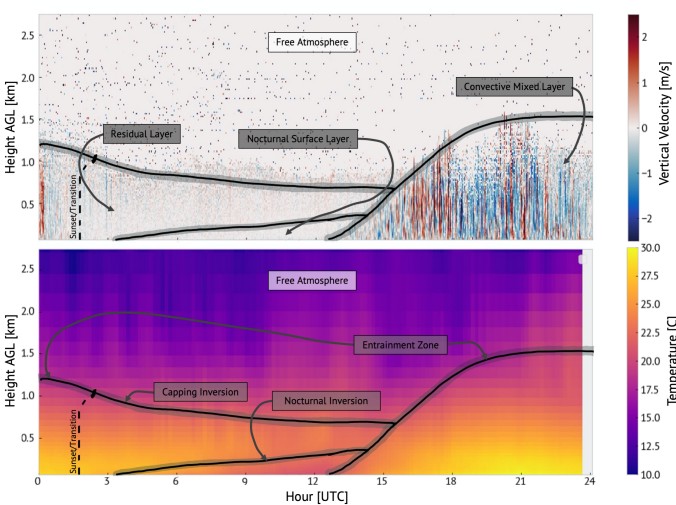

**Figure 1.** A conceptual diagram of the diurnal BL cycle, as often adapted from Stull (1988), overlaid on observed vertical velocity (upper panel) and temperature (lower panel).

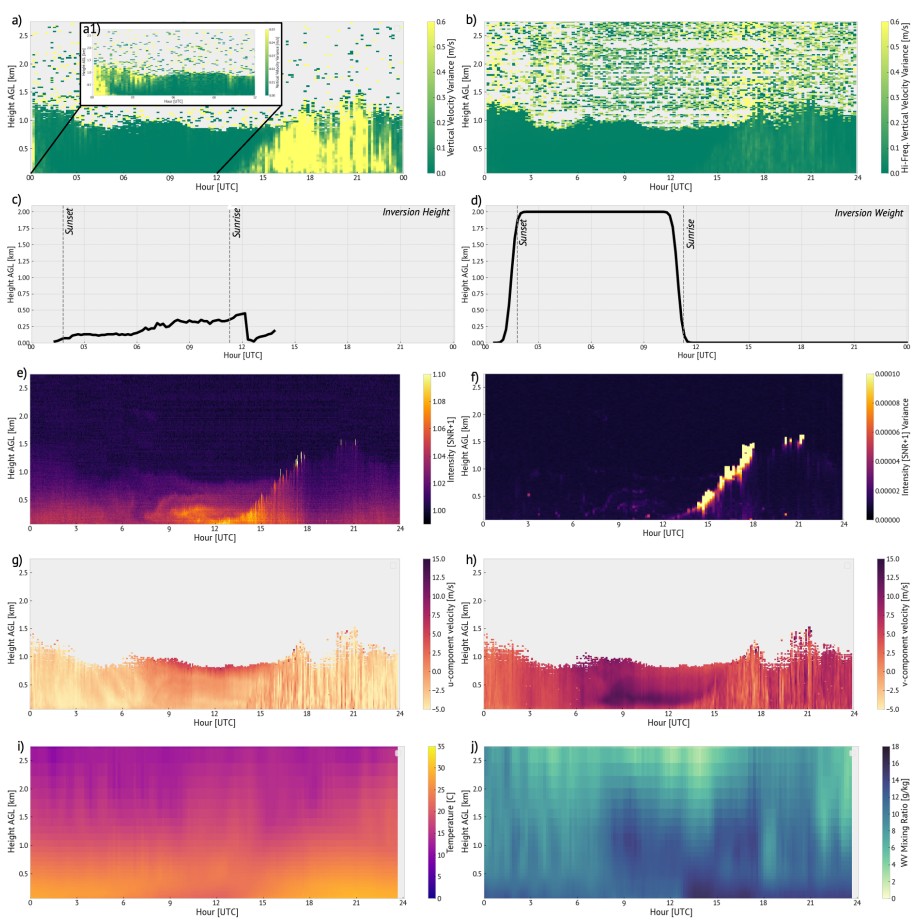

**Figure 2.** CLAMPS1 observations (collected 25 June 2020 in Norman, Oklahoma) used in the fuzzy logic algorithm. Panels (a)–(d) show first generation variables, while panels (e)–(j) show second generation variables. Note that panel (a1) is a subset time window of panel (a) with a more narrow colorbar to highlight overnight vertical velocity variance values. Panel (b) is similar to panel (a) but considers only the high-frequency fluctuations in vertical velocity, as described in the text. Panels (c) and (d) have local sunrise and sunset times marked for reference. Panel (d) is not an observed field, but the weighting applied to inversion height, shown in panel (c).

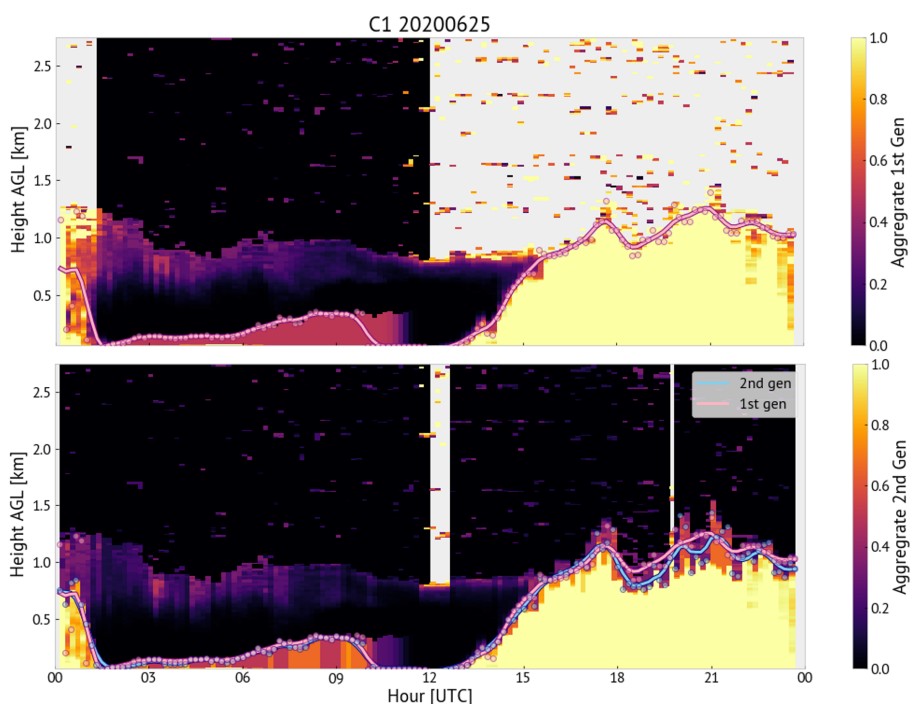

**Figure 3.** Fuzzy logic aggregate fields from the first (upper) and second (lower) generation steps of the algorithm. BL height estimates are traced on top of the color fill aggregate values. Circles show BL height estimates which are computed every 10 minutes, while the curve shows the smoothed estimate after passing through a triangle weighting function.





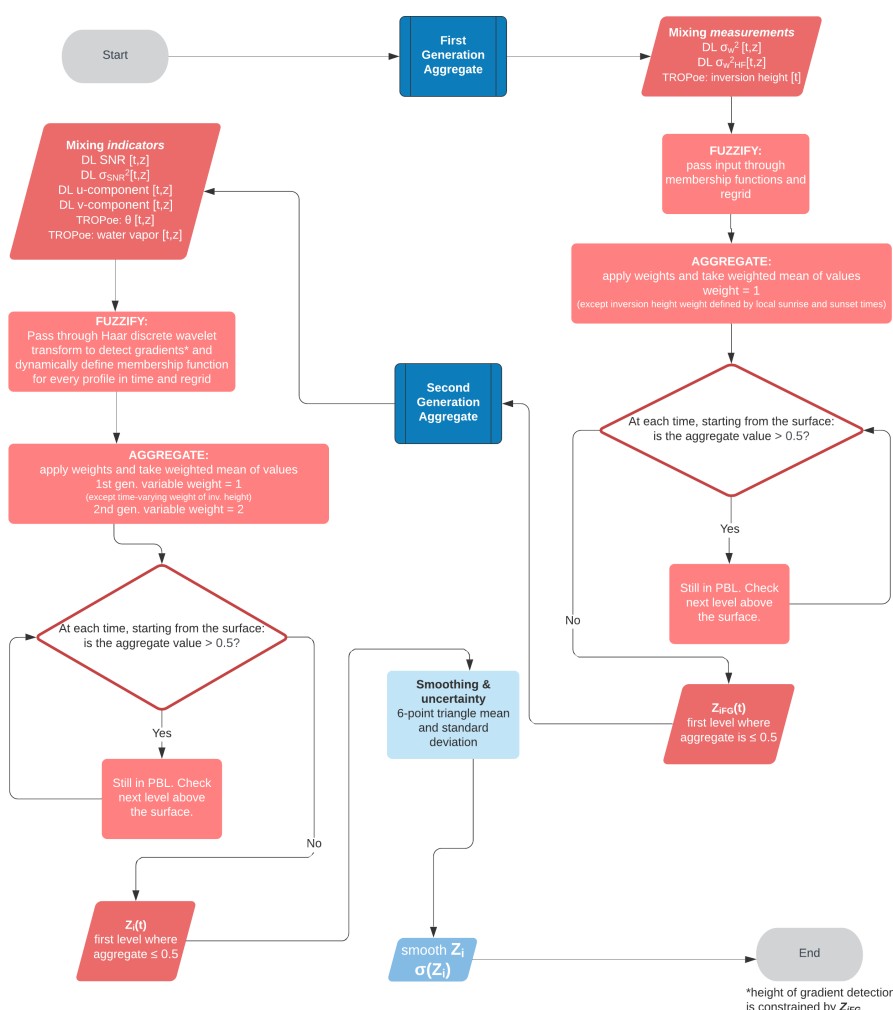

**Figure 4.** A flowchart visualization of the fuzzy logic algorithm steps used to detect boundary-layer height using multiple instruments onboard the CLAMPS platform.



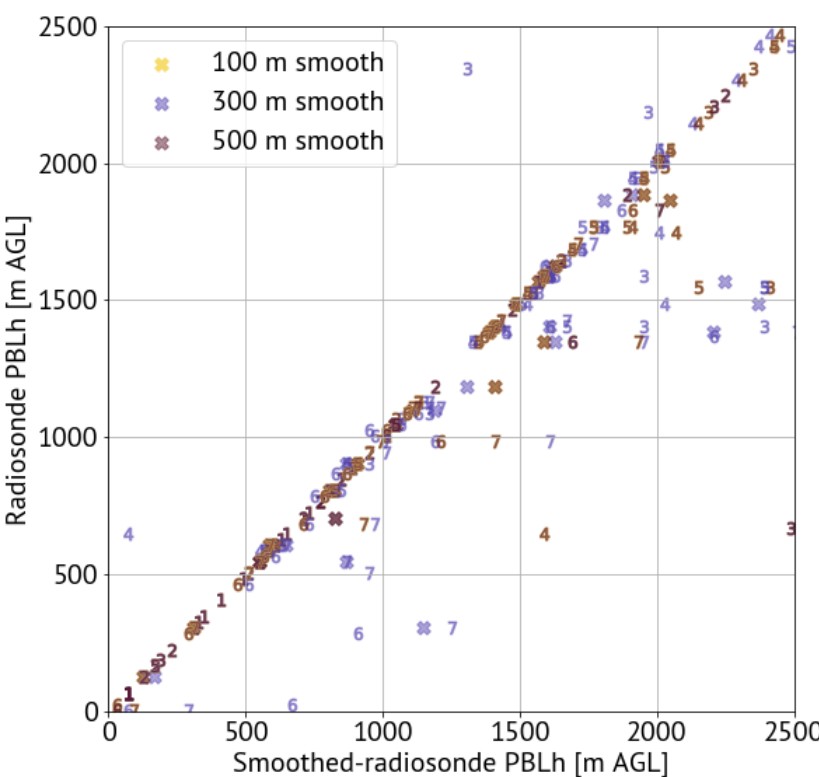

**Figure 5.** CHEESEHEAD sounding BL height estimates on the original vertical resolution are compared with those from when 100-m, 300-m and 500-m Savitzky–Golay smoothing is applied. The marker styles reference BL height estimation technique, where an X marker indicates the median of all methods, and numbered markers reference the methods as they are listed in Table 2.



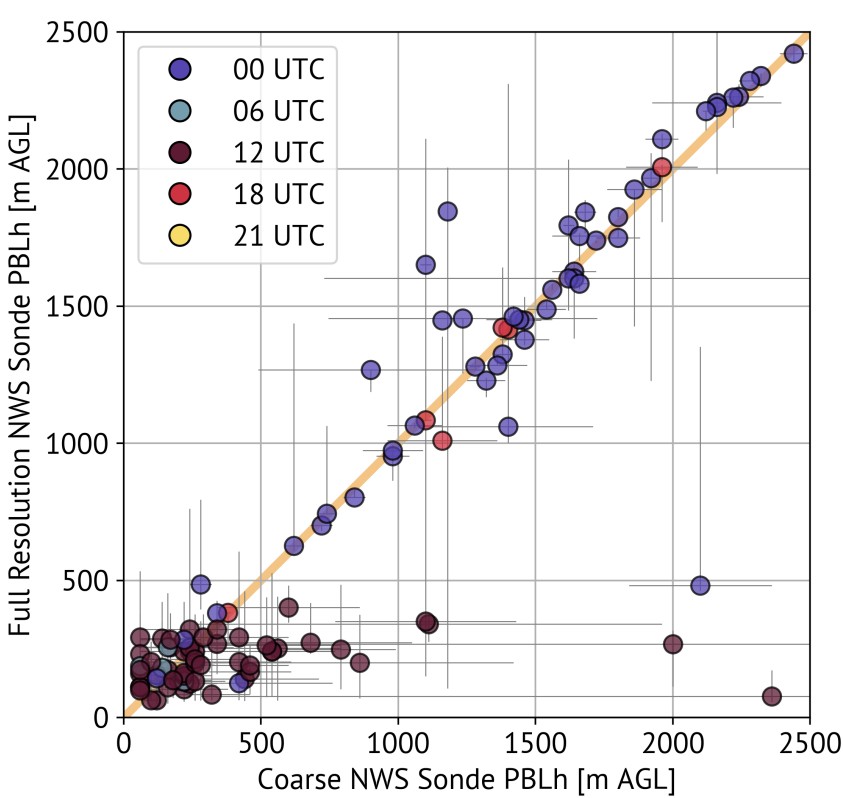

**Figure 6.** Comparison of BL heights derived from coarse, publicly available National Weather Service radiosonde data and BL heights derived via identical methods using the full resolution radiosonde data not available on public archives. Points are color-coded based on nominal launch hour. Error bars (grey) represent the spread of BL heights derived from all methods described in the text as determined from the 25th and 75th percentiles.





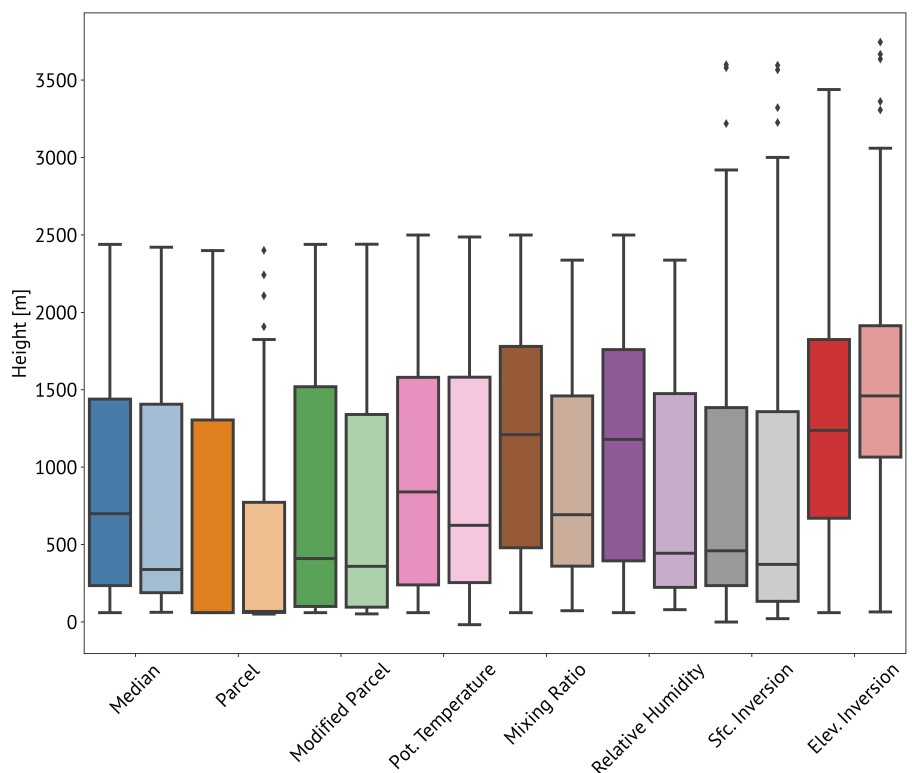

**Figure 7.** Box and whisker diagrams comparing radiosonde-based BL height estimates from coarse (dark colors) and high (light colors) resolution datasets. Only radiosonde profiles for which we had both coarse and high resolution data are included in this analysis. The crossbars mark median values.



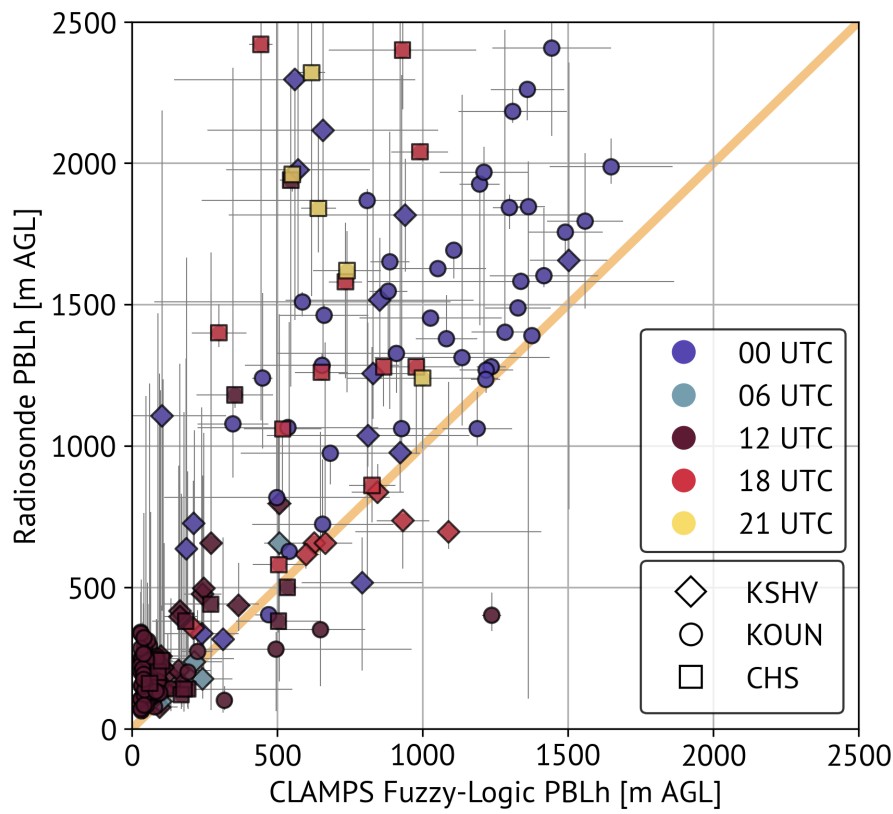

**Figure 8.** Comparison of CLAMPS-derived BL heights and median radiosonde-derived BL heights. Error bars (grey) indicate the range of BL heights detected within ± 30 min and the full range of BL heights from each method for the CLAMPS and radiosonde data, respectively.



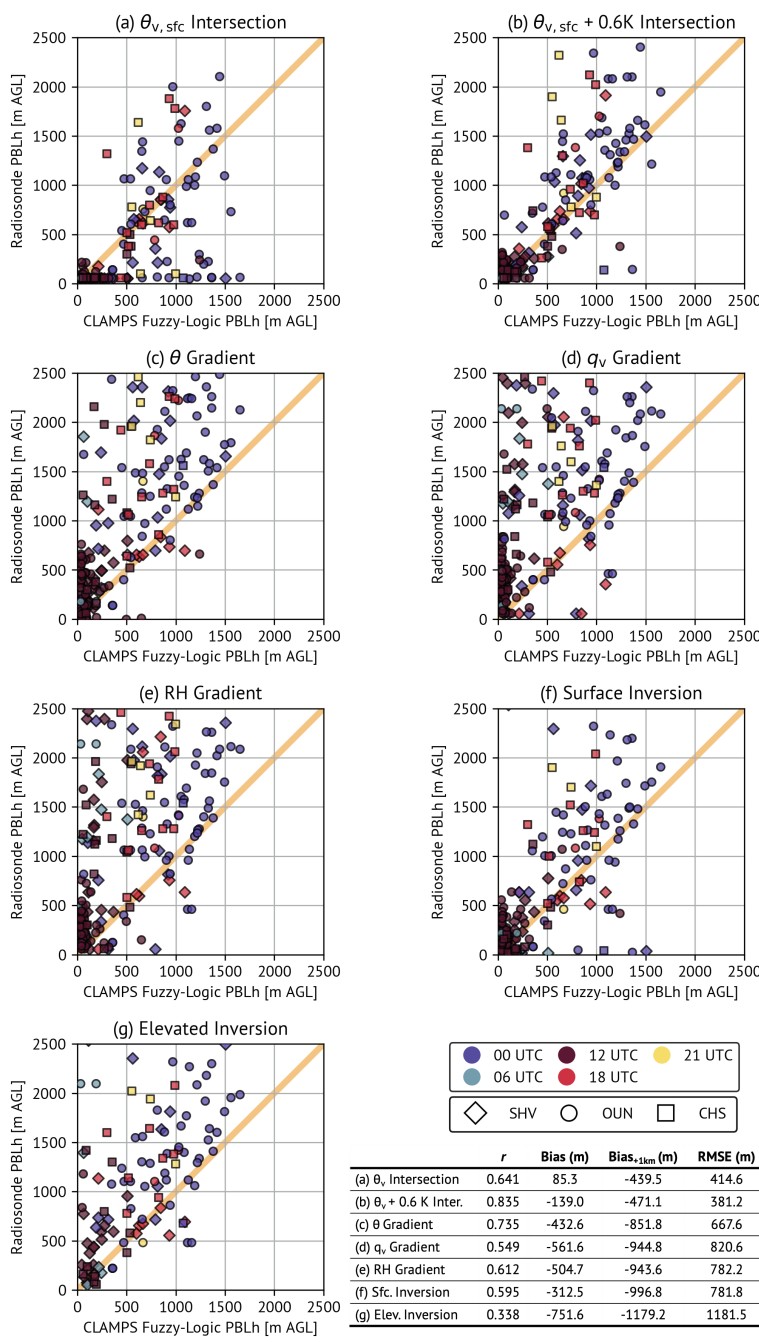

**Figure 9.** As in Fig. 8, but for each of the seven individual BL height detection methods shown in Table 2. Root mean square error (RMSE), $R^2$, and bias (computed for all values and computed only for cases when the BL height is 1 km or greater) for each of the methods is shown in the table.




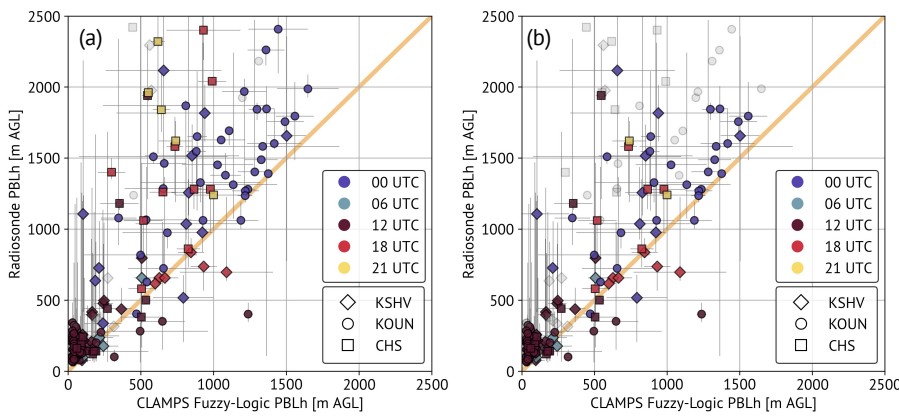

**Figure 10.** As in Fig. 8, but with the exclusion of pairs to which criterion (1) apply (panel a) and the successive exclusion of pairs to which criteria (1) and (2) apply (panel b).





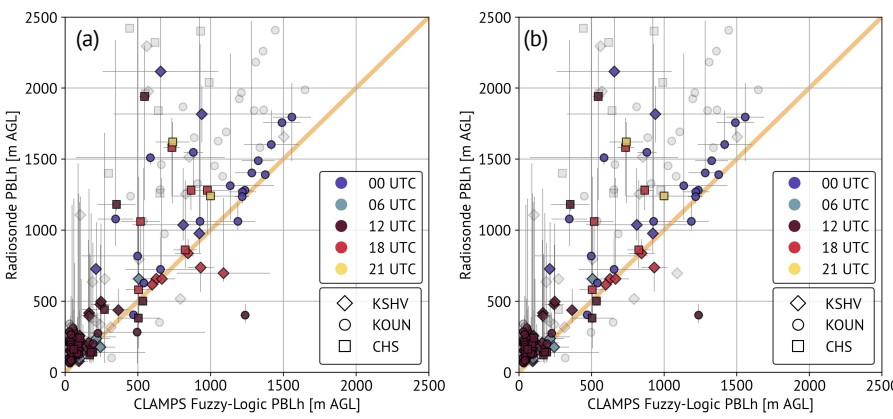

**Figure 11.** As in Fig. 8, but with the successive exclusion of pairs to which criteria (1-3) apply (panel a) and the successive exclusion of pairs to which all criteria (1-4) apply (panel b).



**Table 1.** The core three instruments on board each CLAMPS facility.

| CLAMPS1 | | | CLAMPS2 | | |
|---|---|---|---|---|---|
| **Doppler Lidar** | **AERI** | **MWR** | **Doppler Lidar** | **AERI** | **MWR** |
| Halo Streamline | ABB AERIv4 | RPG HATPRO G4 | Halo Streamline XR+ | ABB AERIv4 | 2021-Current: RGP HATPRO G5 |
| *Upgraded laser (XR equiv.)* | | | *38 m/s Nyquist* | | *2020: None Installed* |
| *38 m/s Nyquist* | | | | | *2016-2019: Radiometrics MP3000A* |



**Table 2.** Methods used to determine BL height from observed radiosonde data. Adapted from Seidel et al. (2010).

| Number | Type | Variable | Method | Reference |
|:---:|:---:|:---:|:---|:---:|
| 1 | parcel | $\theta_v$ | Height where $\theta_v$ profile equal to $\theta_{v,sfc}$ | Holzworth (1964) |
| 2 | methods | $\theta_v$ | Height where $\theta_v$ profile equal to $(\theta_{v,sfc} + 0.6\text{K})$ | Coniglio et al. (2013) |
| 3 | | $\theta$ | Height with maximum $\theta$ gradient | Oke (1988); Stull (1988) |
| 4 | gradient methods | $q_v$ | Height with minimum $q_v$ gradient | Ao et al. (2008) |
| 5 | | RH | Height with minimum RH gradient | Seidel et al. (2010) |
| 6 | inversion | $T$ | Top of surface-based temperature inversion | Bradley et al. (1993) |
| 7 | methods | $T$ | Bottom of elevated temperature inversion | Seidel et al. (2010) |




**Table 3.** Values computed for $r$, bias, and RMSE based on comparisons between the proposed fuzzy logic algorithm BL height estimates and radiosonde-based BL height estimates. The data source provides information about the campaign period and which CLAMPS platform collected observations which were then passed through the fuzzy logic algorithm.

| Data Source | Exclude None | Criteria 1 | Criteria 1+2 | Criteria 1+2+3 | Criteria 1+2+3+4 (All) |
|---|---|---|---|---|---|
| CHEESEHEAD (C1) | 28 | 26 | 19 | 19 | 17 |
| PBLTops/OUN+NWC/RIL (C1) | 73 | 65 | 49 | 44 | 43 |
| PBLTops/OUN (C2) | 21 | 20 | 17 | 12 | 12 |
| PBLTops/SHV (C1) | 47 | 42 | 40 | 34 | 32 |
| Exclusion Percentage | 0 | 10.2 | 27 | 34 | 37.8 |
| r | 0.804 | 0.827 | 0.846 | 0.839 | 0.841 |
| Bias (m) | -294.2 | -261.8 | -181.1 | -160 | -168.6 |
| RMSE (m) | 512.8 | 464.9 | 356.3 | 334.9 | 338.4 |

| | Afternoon/Evening | Overnight/Morning |
|---|---|---|
| Count | 37 | 67 |
| r | 0.618 | 0.525 |
| Bias (m) | -217.7 | -111.7 |
| RMSE (m) | 450.1 | 256.6 |