# Peer review of "A multi-instrument fuzzy logic boundary-layer top detection algorithm"

_EGUsphere, 2023_

## Referee Comment (RC2)

**Review: "A multi-instrument fuzzy logic boundary-layer top detection algorithm"**

This study advances the fuzzy logic methodology initially introduced by Bonin et al. (2018) for the detection of boundary-layer top by integrating data from multiple remote-sensing instrument observations including both kinematic and thermodynamic observations. The research demonstrates that this novel approach yields reasonable estimations of boundary-layer height under both daytime and nocturnal conditions. The utilization of a synergy between multiple instrument measurements is critical for enhancing and ensuring the accuracy of boundary-layer top estimations. Consequently, this study scientifically significant and aligns well with the journal's scope. However, the manuscript is now well organized. Noteworthy concerns are outlined in major comments provided below. Significant revisions are needed before the manuscript be accepted for publication.

**Major comments:**

1. Given the new approach expand upon the work of Bonin et al. (2018), it is essential to include BL estimations by the Bonin et al. (2018) method in the comparisons with radiosonde BL estimations, to show and validate the improvement of the new approach.

2. Figure 1 lower panel: the Entrainment Zone is misleading in the plot. Is the entrainment zone the region between the capping inversion and free atmosphere, with values of ~ 1km?

3. Figure 2: Caption: what are the names and units of the variables in panels (a)-(j)? What's the definition of 'high-frequency' vertical velocity variance? Figure 2 comprises eight panels, but only panels 1-4 were discussed in the text. Why different color schemes are used for different panels. The plots and color schemes make it difficult to see boundary layer structures. What information should reader get from these panels?

4. Page 5 line 154: How the sensitivity of the Haar wavelet dilation was examined and why it is low? From Sawyer and Li (2013), the dilation length is critical to find the signal peaks.

   References:

   Sawyer, V., & Li, Z. (2013). Detection, variations and intercomparison of the planetary boundary layer depth from radiosonde, lidar and infrared spectrometer. *Atmospheric environment*, *79*, 518-528.

5. Page 6 line 161: what physically is P? Is it the peak height or the peak magnitude?

6. Page 8 line 233: Is 'high-resolution' radiosonde refer to vertical or temporal resolution? How frequent was radiosonde launched each day? Similarly, how coarse is the coarse radiosonde data, e.g., what's the resolution?

7. Figure 5: For cases when smoothed-radiosonde PBLH are deeper than radiosonde PBLHT, how to determine/make sure that the deeper values are better/more accurate?

8. Page 9 Line 272-286: This paragraph is confusing. Line 275 states that 'this comparison appears to support the Seidel et al. (2010) findings' that 'high-resolution data can change the estimate in statistically significant ways', while line 285 states that 'there is no indication…using high resolution data yields more accurate BL height values'. Aren't these two statements contradicting with each other?

9. Exclusion of cases were done manually for the four criteria. These criteria are subjective and not well defined. For example, how to determine cases that BLH height was ambiguous (criteria 1)? How to determine at what height CLAMPS observation not reliable to detect PBLH (criteria 2)? When radiosonde and CLAMPS methods identified different parts of the transition region, which one should be trusted (criteria 3)? How to define 'complex/non-canonical BL structures (criteria 4)? Practically, how do users know when to trust or not trust CLAMPS PBLH estimations? Suppose CLAMPS will be run automatically, how to qc label CLAMPS PBLH estimations from the 'bad atmospheric conditions?

10. Are all the comparisons for clear-sky conditions, e.g., no clouds and precipitation?

11. Figure 10 and Figure 11 could be consolidated into a single figure for better clarity and coherence.

12. Figure 11: Even after all exclusions, CLAMPS PBLH is generally still lower than radiosonde PBLH, any speculations of the causes?

**Minor comments:**

1. Page 1 line 19: Given numerous past studies of atmospheric boundary layer and various of BL height estimation methods, it is unrealistic to claim that BL 'yet is also one of the least observed portions of the atmosphere'.

2. Page 2 line 49: it is not clear what does the 'positive impacts' mean.

3. Page 3 line 72: what are buoyancy processes within nocturnal stable boundary layers?

4. Page 3 line 77: Repeating words 'such as the'.

5. Page 5 line 129: what is a 'complete failure to detect a buoyancy-driven BL'? is the 'buoyancy-driven BL' similar as the buoyancy processes within nocturnal stable boundary layers? Under what conditions and how often does the 'complete failure' occur?

6. Page 10 line 303-304: Description of Fig.8 is repeating as in the line 294-295.

---

## Author Comment (AC1)

**Response to reviewers for the paper** *A multi-instrument fuzzy-logic boundary-layer top detection algorithm* **by Elizabeth N Smith and Jacob T Carlin -** *Anonymous Reviewer # 1*

We thank the reviewer for their time and comments on our paper. To guide the review process we have copied the reviewer comments in black, italicized text. Our responses are in regular blue font. We have responded to all the referee comments and, when appropriate to do so, shared the resulting alterations to our paper in blue, italicized text. Any line number references are to those in the originally submitted manuscript.

*This paper presents the results of using a fuzzy logic algorithm with a combination of doppler lidar, radiance [interferometer] and microwave radiometer, with validation done by comparisons with radiosonde data. I feel the paper needs a major revision as the presentation is confusing and the results are not very clear. I realize that there are lots of different cases to present, and many options in the radiosonde comparisons, but in the end, it's not obvious as to how these retrievals might ultimately be used. But I think a rewrite could make it much easier to extract this information. My specific comments are:*

1. *Both the and abstract should state more clearly what observations are being used for the retrieval and validation (see the first sentence above).*
   We modified language in both the abstract and the Summary and Outlook sections to be more explicit about the three instruments used to develop the algorithm, while remaining clear that the algorithm could be adapted to use other similar observation types.
   Abstract: In the second paragraph the first sentence and last sentences were modified:
   *This study introduces a fuzzy-logic algorithm that leverages the synergy of multiple remote-sensing boundary-layer profiling instruments: Doppler lidar, infrared spectrometer, and microwave radiometer.*
   *...*
   *While developed with the three instruments mentioned above, the fuzzy-logic boundary-layer top detection algorithm, called BLISS-FL, could be adapted for other wind and thermodynamic profilers. BLISS-FL is released publicly fostering collaboration and advancement within the research community.*
   Summary and Outlook: In the opening paragraph of this section we added a parenthetical after platforms
   *While this algorithm was developed for use with the CLAMPS platforms (i.e., Doppler lidar, infrared spectrometer, and microwave radiometer), it could be applied to or adapted for similarly instrumented facilities such as but not limited to the Department of Energy's Atmospheric Radiation Measurement profiling facilities.*

2. *Line 110: do you mean the first generation step in B18, or is this changed in the current algorithm?*
   We mean there is an addition to the algorithm at a certain stage, and we will highlight it when we get there. In addition to the existing language "with any deviations from or expansions upon B18 highlighted", we modified the text to more clearly state "*At this stage the algorithm design deviates from the design of B18*" at the beginning of the paragraph which started on line 125 in the original manuscript. This language should make it clearer that the differences are now intentional and by design, not only differences in available variables. We hope that adding this language at the location when the addition to the algorithm is introduced makes it clear that prior to this the algorithm design is generally consistent.

3. *Line 126: Why are buoyant processes important during the night? This seems counter intuitive as buoyancy should grow during the daytime.*
   Buoyancy does grow during the day. Buoyant processes are not limited to the growth of buoyancy. Specifically in the scope of the algorithm we are discussing, the first generation step relies heavily on measures of mixing, or in other words observations showing that mixing is ongoing. In this case, that means mostly mechanical mixing observable by Doppler lidar. Of course, the buoyant processes related to daytime heating can lead to that mixing, but those buoyant processes are not directly observed by the Doppler lidar. Buoyant processes are important during the night when the daytime boundary layer decays and the mechanically driven processes may become less dominant, which in the scope of this algorithm can make the Doppler lidar observations of mixing less informative. As such information about the presence of the daytime capping inversion, thermodynamic structure in the residual layer, and the evolution of any surface inversion is critical at night. In non-canonical cases, the classical "Stull" evolution is not representative of boundary-layer evolution including mechanical or buoyant processes. Including both, specifically through the night, is useful during transitions and during non-canonical nocturnal boundary-layer cases.

4. *Line 131: So you mean that the B18 algorithm was not successful in estimating overnight BL height?*
   The reviewer brings up a fair point that the B18 algorithm is applied throughout the full diurnal cycle. In their paper, they do not offer any assessment or verification at night. In any case, the current phrasing implies that they only applied their method to daytime hours which is inaccurate. We will rephrase this for accuracy.
   *In order to capitalize on the availability of thermodynamic profiles and to improve the capability and robustness of our algorithm during nocturnal hours when the mechanical mixing that backscatter-based instruments (e.g., Doppler lidar) observe can be absent or limited to the residual layer (Schween et al. 2014), the first-generation step also includes temperature inversion height as an input variable (Fig 2c).*
   *Schween, J. H., A. Hirsikko, U. Löhnert, and S. Crewell, 2014: Mixing-layer height*

*retrieval with ceilometer and Doppler lidar: From case studies to long-term assessment. Atmos. Meas. Tech., 7, 3685–3704, https://doi.org/10.5194/amt-7-3685-2014.*

5. *Section 3: Do radiosonde observations measure thermodynamic BLH rather than the ML height? If so, is it really a good comparison?*

   In the scope of this algorithm development, we view mixing and thermodynamic contributions to boundary-layer height as coupled. The algorithm includes measures of mixing as it occurs and indicators mixing has occurred. Thermodynamic BLH parameters can be related to indicators (i.e., mixing has occurred) while MLH would be related to measures (i.e., mixing is occurring). Also, as mentioned in the paper, radiosonde observations are a commonly available and known source of data, which allows the comparison to be made against a dataset that is already integral and 'standard' in many atmospheric data communities.

6. *Line 402: It is fine not to recommend a preferred radiosonde PBLH algorithm, but it would be really helpful to see a plot like*

   In the absence of any additional context, we are treating this comment as if it is related to comment 7 as it seems incomplete on its own. We apologize if our assumption is incorrect.

7. *Figure 3 is really helpful to see the evolution of the PBLH estimate during the course of a day. It would be even better if you could plot the radiosonde estimates of PBLH on top of this. There is lots of discussion of how the algorithm does at different times of the day, but much of this could be clarified with a comparison like this.*

   The strength of an algorithm like the one we present here is we are able to capture or at least make estimates about BL evolution that is not able to be captured in typical radiosonde observations. In our case, we are fortunate to have some additional temporal resolution in the CHEESEHEAD radiosonde dataset (which featured up to 4 radiosonde launches/day) that makes this sort of comparison partially possible. However, the days and regimes in which comparison is available are not fully representative of the comparison we do with the full radiosonde dataset. CHEESEHEAD data are collected in a forest canopy, which perhaps has implications for BL height evolution and BL height detection which are not accounted for in this work. In any case, it is the dataset we have available, so we can include the comparison as requested. We now include a series of four days from the CHEESEHEAD project while the 4-times daily radiosonde releases were occurring. This comparison shows a spread of 'performance,' but generally shows what is expected based on the scatter plot comparisons. Differences are smaller during the early part of the daytime BL. The biggest differences typically occur latest in the afternoon.

   To include this figure in this text, we had to reorganize some of the text in Section 4. Previously the text read as introducing the median soundings scatter plot, describing what it shows, then comparing to the scatter plot of all the sounding methods to describe why the median is important. Now the text reads as introducing the median scatter plot,

moving straight to comparing to scatter plots of all methods to describe why all methods are important, then returning to the median scatter plot to describe what it shows. Then we follow that description with complementary information now available from showing the aggregates from CHEESEHEAD with soundings overlaid with respect to time.

[Figure]

8. *The summary section (5) doesn't really have any concrete conclusions. As you say, it's hard to make a fair comparison since a 100 m difference means something very different depending on the time of day. But plotting the height estimates in comparison to radiosondes as a function of time would really help with this.*

The radiosonde time series plot is now included in the manuscript in response to another comment. In a more direct response to this comment, the Summary and Outlook section, particularly the 4th paragraph, was revised to be more explicit about conclusions and outcomes. We discuss that while there is agreement between radiosondes and the algorithm, there is a pattern of underestimation. We speculate about the reasons behind this including the limitations of remote sensors. We also now more explicitly discuss that exploring errors through comparison with respect to time would require more data, specifically more data around the diurnal cycle to create discrete samples and or develop normalization techniques. The changes appear starting at approximately line 422 in the original manuscript.

*This comparison suggests fairly strong agreement between the techniques, but still with a maintained pattern of the fuzzy logic algorithm applied to CLAMPS observations underestimating BL height compared to the median of radiosonde-based methods. Given the instrumentation types included in the algorithm development and evaluation (Doppler lidar, infrared spectrometer, microwave radiometer), we speculate that even in the best-case comparisons, remote-sensing-based observations are simply more inclined to detect the lowest possible indication of BL height (especially in cases where the BL top itself may be a layer with depth). These types of instruments are also more likely to lose measurement capability and resolution with distance from the surface. These reasons*

*could combine to result in systematically lower estimates of BL height than those provided by in-situ platforms like radiosondes.*

*Unlike some similar algorithms, this approach has the capability to utilize thermodynamic observation information and kinematic observation information (when available) to provide BL height estimates throughout the diurnal cycle. Specific analysis is focused on the early morning and late overnight periods. While some statistics included suggest perhaps even minor improvement compared to the daytime group, this may be a misleading result. For example, a bias of 100 meters means something different when the BL height is O(100 m), which is common in the overnight and morning hours, compared to O(1000 m) BL height values in fully developed daytime and afternoon BL. More data is needed in this comparison to understand the role time of day plays in how the fuzzy logic algorithm behaves. More data could provide the opportunity to build time-dependent samples or to develop temporal normalization approaches to control for this important sensitivity.*

9. *Figure 7: It's not clear what you mean by dark and light colors.*
   In the original manuscript, "dark" and "light" colors referred to the shading/brightness of boxplots for each set of colors (e.g., for the modified parcel method, dark green referred to the results using coarse radiosonde data and light green referred to the results using high-resolution data). We revisited this figure to determine how to present the information more clearly. We came to the conclusion that using different colors for each method and relying on the light/dark shading to show the differences between the box pairs for each method was too confusing. We updated the figure to use a single color scheme for all methods, with the labels along the x-axis describing each method. We also are more specific about using the language of 'pairs' to point out that two boxes go with each method. For all of the pairs the darker shade depicts the results for the given method (or median) using coarse resolution data and the lighter shade shows the results for the same method using high resolution data. We added a legend to the figure to reiterate what

the light and dark shades refer to. The updated figure is shown below.

[Figure]

10. *Table 3 appears to be the primary result of this work. Would it help to include a percent difference along with the absolute difference?*
We have added percent differences at the suggestion of the reviewer. The values generally support the same outcomes. There is a roughly -33% bias overall, decreasing with the first two exclusions. When broken down by time of day the morning bias (~ -42.5%) is roughly twice the afternoon bias (~ -20%).

---

## Author Comment (AC2)

**Response to reviewers for the paper** *A multi-instrument fuzzy-logic boundary-layer top detection algorithm* **by Elizabeth N Smith and Jacob T Carlin -** *Anonymous Reviewer # 2*

We thank the reviewer for their thorough and helpful review of our paper. To guide the review process we have copied the reviewer comments in black, italicized text. Our responses are in regular blue font. We have responded to all the referee comments and, when appropriate to do so, shared the resulting alterations to our paper in blue, italicized text. Any line number references are to those in the originally submitted manuscript.

*This study advances the fuzzy logic methodology initially introduced by Bonin et al. (2018) for the detection of boundary-layer top by integrating data from multiple remote-sensing instrument observations including both kinematic and thermodynamic observations. The research demonstrates that this novel approach yields reasonable estimations of boundary-layer height under both daytime and nocturnal conditions. The utilization of a synergy between multiple instrument measurements is critical for enhancing and ensuring the accuracy of boundary-layer top estimations. Consequently, this study [is] scientifically significant and aligns well with the journal's scope. However, the manuscript is now well organized. Noteworthy concerns are outlined in major comments provided below. Significant revisions are needed before the manuscript be accepted for publication.*

*Major comments:*

1. *Given the new approach expand upon the work of Bonin et al. (2018), it is essential to include BL estimations by the Bonin et al. (2018) method in the comparisons with radiosonde BL estimations, to show and validate the improvement of the new approach.* First, we would like to clarify that our intention is not to *improve* the method that exists in Bonin et al. (2018). We did not mean to convey that we were trying to do so, and tried to avoid language that suggested otherwise. We have revised the text noted by the reviewer to attempt to make it more clear that our intent is to use the B18 framework as a starting point for the development of this algorithm.

   *Bonin et al. (2018) proposed a fuzzy logic algorithm for determining BL height(footnote) from Doppler lidar data and found promising results using data from the Indianapolis Flux Experiment (INFLUX; Davis and Coauthors, 2017). This approach is advantageous in that it combines multiple estimates of BL height, provides a measure of uncertainty for each estimate, and is adaptable to users' individual needs and cases. However, no thermodynamic information is incorporated. In a review of BL height detection approaches, Kotthaus et. al (2023) specifically described the potential benefits of instrument synergy for a variety of applications. We hypothesize that, similar to the suggestions made in Kotthaus et al. (2023), the capability and applicability of BL a height estimation method (in the present case, a fuzzy logic algorithm) could be expanded*

*by incorporating multiple instrument datastreams.*

The method presented in B18 is likely quite appropriate for the deployment in which it was developed, but that deployment isn't entirely representative of other boundary layer observation deployments. Often when deploying instruments to measure in the boundary layer, there can be multiple observation goals which can prevent or at least hinder developing a lidar scan pattern that can depict boundary-layer flow and mixing conditions in such detail (see Bonin et al. 2018; table 2). This is certainly the case for the CLAMPS platforms, which provide the data in this paper. Here we have a small subset of the lidar scan types compared to that of Bonin et al. (2018). For example, their scans include vertical stares, low elevation directional stares to the south and east, RHI scans, and PPI scans at multiple elevations and took about 20 minutes to complete one cycle. Our scan strategy is usually developed for more general observation of the mean wind with faster update times, so it includes a single PPI and a vertical stare. As such a direct comparison between our method and theirs is not possible within our dataset.

2. *Figure 1 lower panel: the Entrainment Zone is misleading in the plot. Is the entrainment zone the region between the capping inversion and free atmosphere, with values of ~ 1km?*

We reviewed figure 1 carefully after receiving this reviewer comment. As mentioned in the caption, this figure is modeled after the well known Stull (1988) diagram of PBL diurnal evolution. In this case, we adapt that evolution diagram by overlaying it on real-world observations and pointing out features defined in the original Stull diagram. Since these are real observations, the features are not *exactly* canonical, but they illustrate the same diurnal cycle nonetheless. These terms and their placements are well established following Stull (1988). We think some of the clarity issues arose from our labeling, particularly in the thermodynamic panel. In the original version, the label for the entrainment zone, which is shown prior to the evening transition and again after the morning transition, was shared across both instances with two arrows. In the new version we split the labels to each instance. We are also explicit about the early evening hours just after 00 UTC including the prior day's entrainment zone. We also added a marker for the morning transition period to help separate the capping inversion and entrainment zone.

Finally, we made all arrow markers larger for clarity. The updated figure is shown below.

[Figure]

3. *Figure 2: Caption: what are the names and units of the variables in panels (a)-(j)? What's the definition of 'high-frequency' vertical velocity variance? Figure 2 comprises eight panels, but only panels 1-4 were discussed in the text. Why different color schemes are used for different panels. The plots and color schemes make it difficult to see boundary layer structures. What information should reader get from these panels?*
It was an oversight to not include references to panels (e)-(j) in the text. They should have been included and have now been added in the description of the second generation step. We also included a more explicit definition for high frequency vertical velocity variance. The colors are different for different variables or variable types. The ranges for the color bars are standardized for comparison to each other (i.e., *u, v*) and to represent typical mid-latitude observed ranges. This multipanel figure, while dense, is intended to offer the reader the opportunity to see the data that is ingested into the algorithm to produce the 1st generation and 2nd generation aggregates and the BL height estimates used in the walk through of the algorithm.

4. *Page 5 line 154: How the sensitivity of the Haar wavelet dilation was examined and why it is low? From Sawyer and Li (2013), the dilation length is critical to find the signal peaks. References: Sawyer, V., & Li, Z. (2013). Detection, variations and intercomparison of the planetary boundary layer depth from radiosonde, lidar and infrared spectrometer. Atmospheric environment, 79, 518-528.*

The sensitivity was examined by trialing different dilations, including the 100 m used in this application, 200 m as used in Bonin et al. 2018, and 300 m. In these trials, there were differences to the wavelet transform magnitude itself. However, applying the peak search and conditions to that wavelet transform led to results that were not sensitive to the dilation.

5. *Page 6 line 161: what physically is P? Is it the peak height or the peak magnitude?*
   We revisited the expression of these conditions to make sure we had them in a clear form. We came to the conclusion that in addition to better defining P we could clarify the whole expression. The equation itself has been rewritten so the notation is improved and the middle condition is expressed more directly. We also revised the text following the expression to more clearly and accurately explain the terms and how/when they result in different values.

$$M(z) = \begin{cases} 1, & z \leq zmin \\ 1 - \frac{\int_{z_{min}}^{z} P(z)dz}{\int_{z_{min}}^{z_{max}} P(z)dz}, & z_{min} < z \leq z_{max} \\ 0, & z > z_{max} \end{cases}$$

*where M is the membership function, P is the profile of peak magnitudes, and z is height above the surface. Since only select peaks are retained, P is nonzero only at the heights of retained peaks. In this notation, $z_{min}$ and $z_{max}$ are the heights where the lowest and highest peaks occur, respectively.*

As is now written in the manuscript P is the vertical profile of peak magnitudes (shown below in blue in the figure below) and is zero everywhere except for at the height of the retained peaks. The new definition for M(z) shows that between the lowest and highest peaks (i.e., the middle condition), the membership function (shown in black below) starts at 1 and decreases at each peak proportional to that peak's magnitude relative to all peaks together. We hope this makes the membership function more clear.

[Figure]

6. *Page 8 line 233: Is 'high-resolution' radiosonde refer to vertical or temporal resolution? How frequent was radiosonde launched each day? Similarly, how coarse is the coarse radiosonde data, e.g., what's the resolution?*
These paragraphs have been edited to include the requested information about resolution and timing.

*Research radiosondes from the CHEESEHEAD campaign (Vaisala research grade radiosondes processed by the National Center for Atmospheric Research; NCAR/EOL In-Situ Sensing Facility and University of Wisconsin Space Science and Engineering Center (SSEC) 2019) and operational National Weather Service radiosondes at OUN from the NWC/RIL and PBLTops campaigns make up the high-resolution radiosonde dataset. These soundings have a mean vertical resolution of approximately 5 m in the lowest 3 km and the exact launch time recorded. Operational radiosondes are typically launched only twice per day for 0000 and 1200 UTC observations. In cases of forecast*

*need, special radiosondes are occasionally launched at other times, usually on a 6-hourly interval. During CHEESEHEAD, radiosondes were nominally launched once daily at 1800 UTC. During two intensive periods the launch frequency increased to four times daily. CLAMPS was only deployed during one of those periods which occurred from 23 September 2019 to 28 September 2019. To prevent erroneous…*

*…*

*At SHV during PBLTops and whenever high-resolution data were otherwise unavailable, coarse-vertical-resolution, publicly available radiosonde observations were used instead. These radiosondes are provided with data recorded at mandatory and significant pressure levels. According to Schwartz and Govett (1992) these levels were determined for a variety of reasons, including but not limited to consistency between existing database conventions at the time of alignment. The resulting vertical resolution varies with height, from O(10) m to O(100) m spacing; 300 m is the average.*

7. *Figure 5: For cases when smoothed-radiosonde PBLH are deeper than radiosonde PBLHT, how to determine/make sure that the deeper values are better/more accurate?*

There is no real measure by which we can say one radiosonde BL height estimate, smoothed or not, is more accurate. In our original approach, the authors visually examined the median and spread of the multiple methods on the suite of CHEESEHEAD soundings to verify that applying smoothing was not introducing unphysical or detrimental estimates. To examine the reviewer's question more carefully, the authors returned to this task. To identify the most impacted cases, for each CHEESEHEAD sounding available the BL heights estimated using unsmoothed data were directly compared to those estimated with the smoothed data. In any case where the difference between unsmoothed and smoothed estimates (from the same technique) was more than ⅓ of the BL height magnitude (represented on the 1-1 line), that sounding case was flagged for closer examination; 10 soundings were identified. Radiosonde BL height estimates were plotted in a similar manner as in figure 5, using color and index numbers to identify the BL height estimation method and sounding (by index number in our datafile), respectively. The blue lines show the PBL height estimate difference threshold boundaries and grey lines show intervals of 250 m.

[Figure]

The soundings that fall outside the boundaries are as follows (with the method resulting in the large difference listed alongside the sounding timestamp):

1: 2019-09-21 18:05:01 — Median
4: 2019-09-25 21:45:03 — PT
9: 2019-09-28 03:00:01 — Sfc, Elev
10: 2019-09-28 21:44:15 — PT, Elev
12: 2019-09-30 18:00:06 — Sfc, Elev
13: 2019-10-01 18:00:48 — Elev
15: 2019-10-03 18:17:36 — Median, Sfc, Elev, RH
17: 2019-10-05 18:01:29 — Median, Sfc, Elev
18: 2019-10-06 18:01:24 — q
21: 2019-10-10 18:00:03 — PT

These soundings were carefully examined manually to compare the unsmoothed profiles to the smoothed profiles and the resulting suite of BL height estimates from each. Most commonly, inversion based methods were noted (ten instances; multiple methods can be noted on a single sounding). Gradient based methods were noted five times, and large shifts in the median were only noted in three instances. In a number of these soundings there are saturated or near-saturated layers below ~2.5 km. In non-saturated cases, the moisture profile is usually still quite moist and characterized by many local maxima and minima. It is possible that these are real local pockets of moistening and drying in shallow layers in the mid boundary layer, but it is unlikely that such shallow layers (especially when there are many) are markers of the boundary layer top. While moisture was a recurring pattern in the soundings, the impacts on boundary-layer height detection methods were not limited to moisture variables. Similar minor, shallow pockets in temperature profiles impacted temperature gradient and inversion-based methods. In some of these cases, smoothing helps the BL height estimation methods which rely on gradients select what looks like a more "reasonable" BL height in the authors' visual assessment. In others, smoothing simply leads to these methods selecting the next most dramatic gradient available in the series of maxima and minima in a noisy profile. In saturated cases, smoothing may not lead to any change, or it may lead to some moisture based methods identifying the top of a cloud layer instead of the bottom (or vice versa). In short, some moisture profiles are difficult to use for characterization of BL height using moisture based methods. It is worth noting that this difficulty, particularly with gradient-based methods, is consistent with the results we note in our full vs coarse resolution comparisons at the end of section 3. Those results were different than the Siedel et al. 2010 findings, but consistent with the more detailed look we have described in this response. Overall, this points to the importance of using the median, which is rarely shifted by much with smoothing. When the median is shifted, it is likely due to outlier adjustment.

8. *Page 9 Line 272-286: This paragraph is confusing. Line 275 states that 'this comparison appears to support the Seidel et al. (2010) findings' that 'high-resolution data can change the estimate in statistically significant ways', while line 285 states that 'there is no indication…using high resolution data yields more accurate BL height values'. Aren't these two statements contradicting with each other?*

The language in this paragraph was a bit confusing, we agree. We made some specific edits to the language around the comparison of Fig 6 and 7 and removed the word error in reference to profile resolution differences to hopefully lead the reader more directly through the subtle differences between what our data and the Seidel et al. (2010) data show. We do agree with Seidel et al (2010)'s finding that high-resolution data can change the estimate in statistically significant ways, but without any real measure of 'truth' we cannot say what is really more accurate. Using the word error when discussing profile resolution differences made that part particularly unclear and we are glad the reviewer

brought this forward for us to fix.

*Seidel et al. (2010) found that while coarse data can be sufficient to detect BL height, the use of high-resolution data can change the estimate in statistically significant ways. A comparison of the medians of radiosonde-derived BL heights in cases where high-resolution and coarse data were available simultaneously is shown in Figure 6. This comparison initially appears to support the Seidel et al. (2010) findings. While many of the afternoon and early evening soundings are close to the one-to-one line, several of the morning soundings fall to the right and below the one-to-one line, suggesting the high-resolution based estimates are lower than the coarse estimates. The 'error' (more accurately, uncertainty) bars, which represent the spread (i.e., interquartile range) of BL heights detected by the seven methods, also appear to cover a wider range for the high-resolution soundings. However, if we further examine the comparison by looking at individual methods separately, we find slightly different results than those in Seidel et al. (2010). Their results showed methods that were computed from the surface up, such as the parcel- and inversion- based methods, were most sensitive to profile resolution, while Fig 7 suggests…*

9. *Exclusion of cases were done manually for the four criteria. These criteria are subjective and not well defined. For example, how to determine cases that BLH height was ambiguous (criteria 1)? How to determine at what height CLAMPS observation not reliable to detect PBLH (criteria 2)? When radiosonde and CLAMPS methods identified different parts of the transition region, which one should be trusted (criteria 3)? How to define 'complex/non-canonical BL structures (criteria 4)? Practically, how do users know when to trust or not trust CLAMPS PBLH estimations? Suppose CLAMPS will be run automatically, how to qc label CLAMPS PBLH estimations from the 'bad atmospheric conditions?*

We describe in the text that each matched pair was manually interrogated for inclusion or exclusion in a similar approach as Banghoff et al. (2018). We do not intend to suggest that these criteria are any sort of objective measures. As we describe in the text our initial comparisons (shown in Figure 8) are made without consideration of the atmospheric conditions at the time of the observations. In this set of comparisons with successive exclusions, we want only to better understand the behavior of the algorithm by eliminating potential 'outside' influences not related to the algorithm itself. The validation of new proposed algorithms using subjective human assessment is a common practice when a known truth is not available. For example, Hubbert et al. (2018) used human experts in convection to evaluate a proposed hydrometeor classification algorithm. Similar to this study, Bianco et al. (2008) used two human experts to evaluate a fuzzy-logic approach for estimating PBL depth. As such, there are not objective criteria to cite for each exclusion condition. For example, if the authors with experience identifying PBL height from radiosondes could not themselves confidently identify a

PBL height for Criterias 1 and 4, it was deemed ambiguous or complex and not a good case for evaluating algorithm performance. For Criteria 2, if the PBL depth increased steadily and then suddenly leveled off despite continued mixing coinciding with a threshold value of scatterers, it suggests a limitation in how far the lidar signal could reach. Finally, for Criteria 3, it is not our intent to say which estimate is more trustworthy or representative. In fact, the fact that we cannot is the reason we elect to remove those cases from the database in the first place. We aimed to be transparent in this successive elimination by including both the results with all data points and those with low-confidence cases (as judged by human experts) excluded, and by having each author perform an independent assessment of each of the four criteria. Despite these exclusions, we still retained nearly two-thirds of the cases originally included. To make the criteria themselves more clear, we have amended their descriptions some, providing a few examples in some cases. We have set them inline in a numbered list in the manuscript.

*In order to better understand the potential causes of discrepancies between the radiosonde and CLAMPS-based BL height estimates and ensure a robust comparison, each matched pair was manually interrogated for inclusion or exclusion in a similar approach as Banghoff et al. (2018). In our case, criteria for exclusion were developed to describe instances in which discrepancies between CLAMPS and radiosonde BL heights may not necessarily reflect the intrinsic performance of the proposed algorithm. These criteria were as follows:*
*1) ambiguous cases where the BL height was unable to be confidently determined from the radiosonde data as multiple methods failed to identify BL height at all or identified levels that upon visual inspection were likely related to other structures (e.g., residual layer, clouds, etc.);*
*2) cases where CLAMPS observations were not able to be collected over a deep enough layer to capture the likely full depth of the BL (primarily DL observations due to a lack of scatterers);*
*3) cases where the BL top (e.g., entrainment layer or capping inversion) was deep and the radiosonde and CLAMPS methods identified different parts of this transition region;*
*4) cases with complex/non-canonical BL structures (e.g., multiple inversions, low-level cloud layers, etc.).*
*To identify cases for exclusion, both authors independently evaluated each time-matched CLAMPS and radiosonde profile to determine whether one or more of these criteria were met without respect to how their BL height estimates compared, then discussed and reconciled any differences. These criteria were used to successively exclude data pairs and get a better sense of algorithm performance. As each criteria was applied successively, more pairs were removed from the comparison dataset as summarized in Table 3.*

Given this paper is introducing an algorithm and not a dataset from CLAMPS, the latter part of the comment is outside the scope of the present work. Any retrieval method or algorithm is only as good as the data it is provided. It is the responsibility of the user to know what they provide to an algorithm. Note that many of the standard accepted radiosonde-based BL height estimation methods (e.g., finding where the surface potential temperature intersects the profile) also rely on assumptions of a canonical BL (i.e., those with surface-up mixing processes) and could similarly fail in non-canonical BLs. In the case of providing it as a dataset, we already provide one quasi measure of confidence in the estimate in the form of the standard deviation over the moving windows, which makes the assumption that very variable BL height estimates are less confident or more uncertain. Regarding QC, that step should be done to data likely before it is provided to the algorithm. Theoretically flags could be passed through the algorithm from the datasets it ingests in a real-time setting, but this use case has not been developed at this time. The criteria discussed here would not be applied as QC criteria as they are not meant to be QC criteria. They are meant to be subjectively applied elimination criteria to provide us a best-case dataset for comparison, eliminating outside impacts on potential algorithm performance.

Bianco, L., J. M. Wilczak, and A. B. White, 2008: Convective boundary layer depth estimation from wind profilers: Statistical comparison between an automated algorithm and expert estimations. *J. Oceanic Atmos. Tech.,* **25**, 1397-1413. doi:10.1175/2008JTECHA981.1.

Hubbert, J. C., J. W. Wilson, T. M. Weckwerth, S. M. Ellis, M. Dixon, and E. Loew, 2018: S-Pol's Polarimetric Data Reveal Detailed Storm Features (And Insect Behavior). *Bull. Amer. Met. Soc.,* **99**, 2045-2060. doi:10.1175/BAMS-D-17-0317.1.

10. *Are all the comparisons for clear-sky conditions, e.g., no clouds and precipitation?*
    All comparisons are in precipitation free periods (over the profilers). The algorithm requires Doppler lidar data and thermodynamic profiler data to provide a boundary layer height estimate. In our case, we are using the AERI instrument for thermodynamic profiles. This instrument does not collect data during precipitation. It detects precipitation and closes a protective hatch to keep its detector dry and safe. Cloudy conditions were not intentionally excluded. It is true that some clouds could have an impact on the algorithm (e.g., premature extinction of lidar signal, radiative impacts on the AERI measurement). However, there are reasons why we don't exclude clouds. First, deciding the threshold for 'what is a cloud' based on the types of observations we have available to us is non-trivial. Second, at times the formation of clouds and their positions can be related to the boundary layer depth itself. Without an independent measurement of cloud conditions, we did not feel confident in creating an objective exclusion criteria.

11. *Figure 10 and Figure 11 could be consolidated into a single figure for better clarity and coherence.*

These panels have been combined into a single figure.

12. *Figure 11: Even after all exclusions, CLAMPS PBLH is generally still lower than radiosonde PBLH, any speculations of the causes?*

In response to this comment and another reviewer comments, revisions were made to the Summary and Outlook section, particularly to the fourth paragraph, that address this comment. It is now more explicit about conclusions and outcomes. We discuss that while there is agreement between radiosondes and the algorithm, there is a pattern of underestimation. We speculate about the reasons behind this including the limitations of remote sensors (capabilities and resolution). We also now more explicitly discuss that exploring errors through comparison with respect to time would require more data, specifically more data around the diurnal cycle to create discrete samples and or develop normalization techniques.

*Minor comments:*

1. *Page 1 line 19: Given numerous past studies of atmospheric boundary layer and various of BL height estimation methods, it is unrealistic to claim that BL 'yet is also one of the least observed portions of the atmosphere'.*

These authors argue that it is realistic. Many of these observation studies are just that: studies. In general the boundary-layer remains routinely under-observed due to a lack of widely available affordable, maintainable, deployable solutions capable of capturing the atmospheric conditions between the surface and hundreds to a few thousand meters above it, where remote systems such as satellites and radars can be more successful at monitoring. This is even more of an issue over Earth's oceans, though this is not discussed in the present study. See Bell et al, 2020, NASEM 2018a, NASEM 2018b, NRC 2010, NRC 2009 (all cited within this manuscript) for more discussion. To make the distinction more clear we have added the word routinely to the paper in the same fashion it is used in this response.

2. *Page 2 line 49: it is not clear what does the 'positive impacts' mean.*

We rephrased this sentence for clarity.
*Recently Tangborn et al. (2021) found the accuracy of the temperature and wind fields in the simulated afternoon convective BL to be improved compared to radiosonde observations when assimilating observations of BL height.*

3. *Page 3 line 72: what are buoyancy processes within nocturnal stable boundary layers?*

Buoyant processes are not limited to the growth of buoyancy. Specifically in the scope of the algorithm we are discussing, the first generation step relies heavily on measures of mixing, or in other words observations showing that mixing is ongoing. In this case, that means mostly mechanical mixing observable by Doppler lidar. Of course, the buoyant processes related to daytime heating can lead to that mixing, but those buoyant processes

are not directly observed by the Doppler lidar. Buoyant processes are important during the night when the daytime boundary layer decays and the mechanically driven processes can become less dominant, which in the scope of this algorithm may make the Doppler lidar observations of mixing less informative. As such, information about the presence of the daytime capping inversion, thermodynamic structure in the residual layer, and the evolution of any surface inversion is critical at night. In non-canonical cases, the classical "Stull" evolution is often not representative of boundary-layer evolution including mechanical or buoyant processes. Including both is useful during transitions and during non-canonical nocturnal boundary-layer cases.

4. *Page 3 line 77: Repeating words 'such as the'.*
   We have made this correction.

5. *Page 5 line 129: what is a 'complete failure to detect a buoyancy-driven BL'? is the 'buoyancy-driven BL' similar as the buoyancy processes within nocturnal stable boundary layers? Under what conditions and how often does the 'complete failure' occur?*
   A failure to detect a buoyancy-driven BL would be the event in which there are no mechanical processes for the algorithm to identify so the algorithm fails, incorrectly, to identify the BL height while buoyant processes dominate. This then carries forward, as this initial estimate is used to constrain which peaks are retained for refining the estimate using the thermodynamic information. We did not examine the frequency, but this could occur anytime there is an absence of measurable mechanical mixing for the algorithm to detect as a measure of mixing ongoing. This is certainly possible at night. We modified the language in the paper to make this a bit clearer.

   *Buoyant processes also play a role in BL development, and are often a dominant process at night when stable boundary layers are more common. In later steps the BL height estimate from the first-generation step is used as a type of constraint on the second-generation step. Thus if there is a complete failure to detect a BL height (presumably in situations in which the BL depth is driven by buoyant instead of mechanical processes) in the first-generation step, the second-generation step is unable to recover.*

6. *Page 10 line 303-304: Description of Fig.8 is repeating as in the line 294-295.*
   This language was intended to draw a comparison between the two figures, not be a repetition of information. We think this helps the reader along in interpreting the meaning of the multiple panels on figure 9, since they are the same analysis as figure 8, just with different data subsets. However, after fulfilling some requests from other reviewer comments this particular section of writing has been reorganized. In the revision there is less repetition naturally, but we also took care to make the comparison between the figures without repeating the description of either.

---

## Referee Report (RR1)

**Review: "A multi-instrument fuzzy logic boundary-layer top detection algorithm"**

The revised manuscript improved a lot. However, there are still several noteworthy concerns as outlined in comments below. All line and page numbers are based on the tracked revised manuscript.

**Major comments:**

1. How do the authors know the new algorithm, i.e., the second-generation step, improves boundary layer height estimation, compared with the first-generation step? It should be noted that including low-quality data or including non-critical data could jeopardize BL estimates. For example, in Fig. 3, the second-generation algorithm estimated BL height is almost the same as the first-generation ones during the nighttime, only show some differences during the daytime. However, from the description between line 135-139, it indicates that the first-generation estimation is expected to work well during the daytime, but has issues during the nighttime, which is contradictory to what Fig. 4 shows. The new added figure, Fig. 10, shows the same that the second-generation BL height is almost as the first-generation BL height during the nighttime. Fig. 10 b) (top right) even shows that the first-generation BL heights are closer to sonde-estimated BL height at ~18 UTC and 22 UTC. For Fig. 8, it might be better to include the comparison between the first-generation BL heights and sonde-estimated BL heights.

2. The caption of Fig. 2 is still not clear. The caption should be able to roughly explain the figure so that readers do not need to read carefully in the text to understand the figure. For example, what are first- and second- generation variables? Those need to be clear in the caption. Fig. 2c and line 144, how is the temperature inversion height derived from the first-generation step as it only calculates the vertical wind variance? Fig. 2d, is it the inversion height or weighting function? 'inversion weight' is not defined in the text.

3. Line 139-141: the logic is confusing here. It is indicated that a complete failure to detect BL height in the first-generation step occurs when the 'buoyant processes' dominates. If the first-generation step fails, the second-generation step is 'unable to recover'. However, according to the manuscript, the main advancement of the second-generation step is to improve 'buoyant processes' dominated (nighttime) BL height estimates. The question is that if the first-generation step failed and the second-generation is 'unable to recover', how does the second-generation step improve BL height estimates?

4. The terminologies 'buoyant processes' and 'mechanical processes' are confusing. What exactly are 'buoyant processes' and what exactly are 'mechanical processes' in BL? Any references using these terminologies? In BL, we often use 'shear- or buoyant- driven' turbulence.

5. The manuscript states that 'the Harr wavelet transform is not sensitive to the dilation', which contradict with previous studies (e.g., Sawyer and Li, 2013). Why physically 'the

Harr wavelet transform is not sensitive to the dilation' in this case? If it is not sensitive to the dilation length, why choose to use 100 m instead of, for example, 50 m or 300 m?

6. The author's reply to my comment 6: '*The resulting vertical resolution varies with height, from O(10) m to O(100) m spacing; 300 m is the average.*' It is not clear what does 'O(10) m' and 'O (100) m' mean. How come the average is 300 m?

7. Line 315: From Fig. 6 and Fig. 7, it is not sufficient get the assertion that 'there is no indication from this analysis that using high-resolution data necessarily yields more accurate BL height values'. 1) Fig. 6 shows that the differences between high- and low-resolution mainly occurs at small BL height values., e.g., during time. While Fig.7 shows the absolute BL height ranges for all cases. Small BL heights generally have smaller absolute BL height estimation ranges (across different methods), which large BL heights generally have large absolute BL height estimation ranges. Therefore, Fig. 7 is biased by the large BL heights. 2) there is a clear trend that High-res BL height estimations are much smaller than coarse-res BL height estimations for majority of the methods, as show in Fig.7. Therefore, it is hard to believe that 'coarse-resolution data' has neglected impacts on BL estimations.

8. For the answers to my comment #9: These conditions are also the major reason for many other BL estimates fail too. How does the first-generation BL estimates compare with sonde BL estimates if these conditions were excluded? The core question is still that: how does the author prove that including other data/measurements improves the BL estimation?

9. For the answers to my comment #10: 'deciding the threshold for 'what is a cloud' … is non-trivial'. Can't the DL backscatter intensity be used for cloud detection?

---

## Referee Report (RR2)

**Review: "A multi-instrument fuzzy logic boundary-layer top detection algorithm"**

The authors' reply clarifies some of my comments. However, there still some confusing statements/definitions/discussions for me as listed below.

**Comments:**

1. While I concur that leveraging the synergy of multi-remote sensing boundary layer profiling measurements could enhance PBLH estimations by providing more constraining information, I find the authors' claim that 'the proposed method enables accurate boundary-layer height estimation both during daytime and nocturnal conditions' unconvincing based on the comparisons in Figures 8-11. The PBLHs estimated by remote-sensing (Fuzzy-logic) are consistently lower than those derived from radiosondes for PBLHs > 500 m or during afternoon convective boundary layer conditions (Figure 10). Why or what causes the underestimations from Fussy-logic PBLHs?

2. Based on Figure 2, it appears that the second-generation step yields even lower PBLHs than the first-generation step. This suggests that the second-generation step further underestimates PBLHs under conditions of a convective boundary layer. Is that correct?

3. The comparisons between Fuzzy-logic and radiosonde PBLHs do not seem as robust as those shown in previous studies that exclusively used Doppler lidar measurements, such as those in Tucker et al. (2009) and Krishnamurthy et al. (2021). Could you provide the correlation coefficients between Fuzzy-logic PBLHs and Radiosonde PBLHs as depicted in Figures 8, 9, and 11?

    References:

    Tucker, S. C., C. J. Senff, A. M. Weickmann, W. A. Brewer, R. M. Banta, S. P. Sandberg, D. C. Law, and R. M. Hardesty, 2009: Doppler Lidar Estimation of Mixing Height Using Turbulence, Shear, and Aerosol Profiles. *J. Atmos. Oceanic Technol.*, **26**, 673–688, https://doi.org/10.1175/2008JTECHA1157.1.

    Krishnamurthy, R., Newsom, R. K., Berg, L. K., Xiao, H., Ma, P.-L., and Turner, D. D.: On the estimation of boundary layer heights: a machine learning approach, Atmos. Meas. Tech., 14, 4403–4424, https://doi.org/10.5194/amt-14-4403-2021, 2021.

4. The authors have noted that in the first-generation step, the temperature inversion height was used as an input. Could you clarify how the temperature inversion height is derived? What degree of temperature difference is considered to constitute an inversion? The term 'temperature inversion height' is introduced in line 139 without an explanation of how it is obtained. In the response, the authors claim, *'We have never stated that inversion height is derived from the first-generation step*.' If so, could you specify when the 'inversion height' is derived?

5. Typically, a figure caption should provide a brief explanation of what the figure represents, such as the observations or variables depicted in Figure 2. This would provide readers with basic information, and I don't believe it would be as 'extensive' as the authors suggested in their response. They could relocate the detailed discussions or illustrations of the figure to the main body of the text.

6. Despite several rounds of discussion, the precise definitions of 'buoyancy-driven processes' and 'mechanically-driven processes' remain unclear.

7. The authors replied that '*if dz = the spacing between any two levels in a profile, and we know that dz varies throughout the profile, 300 is the average of all dz values in the profile. The minimum dz that occurs in the lowest levels of the atmosphere is on the order of 10s of meters. The maximum dz that occurs higher up in the atmosphere is on the order of 100s of meters.*' If a group of numbers have a minimum of 10 and maximum of 100, how can the average to be 300?

8. As for the comparison of high- vs. low-resolution sounding data, the authors state, '*Analyzing the methods separately highlights that using high-resolution sounding data tends to lead to lower median BL height, but the ranges and interquartile range values do not change much between coarse and high resolution datasets. There is no indication from this analysis that using high resolution data necessarily yields more accurate BL height values*'. It appears that the authors are suggesting that because 'the coarse and high resolution datasets have similar PBLH ranges and interquartile range values', therefore, '*There is no indication from this analysis that using high resolution data necessarily yields more accurate BL height values*'. I don't believe this is a solid conclusion. For instance, under an extreme condition where the coarse sounding data produce identical PBLHs, resulting in ranges and interquartile range values of zero, it certainly doesn't imply that coarse sounding data provides accurate PBLH estimations.

---

## Author Response (AR2)

**Response to reviewers for the paper** *A multi-instrument fuzzy-logic boundary-layer top detection algorithm* **by Elizabeth N Smith and Jacob T Carlin -** *Anonymous Reviewer # 2*

We thank the reviewer for their continued review of our paper. To guide the review process we have copied the reviewer comments in black, italicized text. Our responses are in regular blue font. We have responded to all the referee comments and, when appropriate to do so, shared the resulting alterations to our paper in a darker blue, italicized text. At times we include additional text from the manuscript in italicized grey text [the marker (TR) refers to tracked revised manuscript].

*The revised manuscript improved a lot. However, there are still several noteworthy concerns as outlined in comments below. All line and page numbers are based on the tracked revised manuscript. Major comments:*

*Major comments:*

1. *How do the authors know the new algorithm, i.e., the second-generation step, improves boundary layer height estimation, compared with the first-generation step? It should be noted that including low-quality data or including non-critical data could jeopardize BL estimates. For example, in Fig. 3, the second-generation algorithm estimated BL height is almost the same as the first-generation ones during the nighttime, only show some differences during the daytime. However, from the description between line 135-139, it indicates that the first-generation estimation is expected to work well during the daytime, but has issues during the nighttime, which is contradictory to what Fig. 4 shows. The new added figure, Fig. 10, shows the same that the second-generation BL height is almost as the first-generation BL height during the nighttime. Fig. 10 b) (top right) even shows that the first-generation BL heights are closer to sonde-estimated BL height at ~18 UTC and 22 UTC. For Fig. 8, it might be better to include the comparison between the first generation BL heights and sonde-estimated BL heights.*

   In order to best respond to this comment, we will break down the concepts addressed in it into two main parts. First there is the general question: How do we know that the new algorithm (i.e., the second generation step) improves boundary-layer height estimation compared with the first generation step? In some ways we could consider some of the reviewer's question (specifically about including low-quality data or non-critical data) to imply that if the second generation step could make the estimate worse, we should not embark on it, and then it follows that only lidar measured vertical velocity matters to this problem and adding any other information cannot help the process. Given the premise of our paper and the content of our literature review, we do not agree with this framing and instead we align with the notion that muli-instrument approaches offer the opportunity to add benefits to one another especially when some of the included instruments experience observation weaknesses from time-to-time. In this framing, we could say that lidar-only

methods of boundary-layer detection may be limited---for example during the nighttime or very stable periods when mechanically-driven mixing processes are unlikely to present much if any target by which the lidar could detect boundary-layer height, or during periods of fog or low cloud when the lidar signal is partially or completely extinguished by water droplets. Including additional instrumentation that can gather other information about the structure of the boundary layer which can be used to estimate the boundary-layer height is obviously beneficial. In a fuzzy-logic approach, which is what we use, it is important to understand the data sources/types (and the quality assurance measures that the user has hopefully applied or understands have been applied to them) that contribute to the result and assign membership functions, weights, and other parameters or controls accordingly to control for any 'issues' with 'low-quality' or 'non-critical' data. Those functions, weights, etc. should be appropriately assigned to allow the most impactful or direct datasets to contribute more. This is what our paper describes. Still, to make the point abundantly clear in the paper, we have added language to the end of the fuzzy-logic algorithm section, immediately prior to the data section.

*In the case shown in Fig. 2 (and in the CLAMPS datasets described next in Section 3), care was taken to understand how the instruments operated, any quality assurance applied to the data during their initial collection and production, and to determine if any additional quality assurance was needed before providing the data to the fuzzy logic algorithm. The CLAMPS platform happens to be a highly automated system which applies a high level of quality assurance automatically, reducing (but not removing) user needs to further quality assure CLAMPS datasets. This may not always be the case, and users must understand the potential impacts data quality may or may not have when providing inputs to any algorithm. In the case of our fuzzy logic approach, the physically-based definition of membership functions limits in the first generation step and the BL height-range constraint in the second generation step both provide some safeguard against entirely artificial or problematic data, but ultimately users must consider the data quality provided to the algorithm when weighing the quality of BL height estimates.*

The second part of this comment includes specific discussion of figures. In the comment, the reviewer mentions figure 4, which for us is the flowchart figure. We assume the reviewer meant figure 3 and are responding as if that is the case. Here the reviewer notes that the first and second generation estimates of boundary-layer height are similar/the same overnight and only different during the day. This is framed as being inconsistent with text on, e.g., lines 135-139 where we state that first-generation estimation is expected to work well during the daytime, but has issues during the nighttime. For completeness the text from those lines are reproduced here.

(TR) *Lines 135-139: "Since the first-generation estimate depends only on measures of*

*mixing, it relies only on mechanically induced turbulence and mixing to determine BL height. Buoyant processes also play a role in BL development, and are often a dominant process at night when stable boundary layers are more common. In later steps the BL height estimate from the first-generation step is used as a type of constraint on the second-generation step.*

Here we explain that the first generation step, as it exists at this point in the paper (and in line with the B18 approach) relies only on mechanically induced processes. We have not introduced any change to the B18 algorithm other than the statement "At this stage the algorithm design deviates from the design of B18." We bring up the potential problem of only relying on lidar data for boundary layer identification through the full diurnal cycle (e.g., buoyant processes) and how such a problem may have effects later in the algorithm (because the 1st gen estimate serves as a constraint in the 2nd gen). We then expand on this explanation, which also explains the behavior seen in the figure.

*(TR) Lines 139-150 Thus if there is a complete failure to detect a buoyancy-driven BL height in the BL height (presumably in situations in which the BL depth is driven by buoyant instead of mechanical processes) in the first-generation step, the second-generation step is unable to recover. In order to capitalize on the availability of thermodynamic profiles and extend the capability to improve the capability and robustness of our algorithm during nocturnal hours when the mechanical mixing that backscatter-based instruments (e.g., Doppler lidar) observe can be absent or limited to the residual layer (Schween et al., 2014), the first-generation step also includes temperature inversion height as an input variable (Fig 2c). as an input variable (N.B.: this is not a measure of mixing). To limit the effect of this additional input beyond periods where buoyancy is more likely to dominate mechanical generation of mixing (e.g., nocturnal stable periods), a time-dependent weighting function is defined based on the local sunrise and sunset time (Fig 2d). This weighting function allows the inversion height to have an effect on the algorithm during the overnight hours with sloped increasing and decreasing weights during the evening and morning transitions, respectively. The membership function in this case is step wise, and thus not authentically a fuzzified field. All levels above and below the inversion height are assigned membership values of zero and one, respectively."*

In other words, inversion height is added as an input during the first generation step. This is the deviation from the B18 approach. Without it, for the case in which no boundary-layer height was detected in the first-generation, the second-generation would have no basis from which to start (since it is required to work within a constrained distance of the first generation estimate). We know that the inversion height is really most important when mechanically-driven mixing processes are minimized/buoyant processes are dominant and this is the time the lidar detection methods are most likely to fail. We expect this to most often occur at night, so we assign a weighting function that only allows this first generation inversion height input to have an impact during the nighttime

hours. During the second generation step, thermodynamic profiles are used at all times of the day. Given that overnight first generation estimates are often dominated by inversion height (i.e., based on thermodynamic profiles), it makes sense that adding more information from the thermodynamic profilers in particular at night would not lead to much change to the estimate in the second generation during that period. This same reasoning can be applied in the figure 10 panel the reviewer cites.  In regard to the upper right hand panel of figure 10: this is a single case that we added on reviewer request as one of a set of examples. On any one day at any one observation time, we could choose to show a 'better' or 'worse' comparison to either generation estimate. We also point out that for this particular 22Z sounding there is a large degree of spread among sounding estimation methods. Forming arguments or opinions about data from a single comparison point is not a particularly informative method and not the method we choose to determine the 'performance' of any technique. Comparing with the first generation estimate is not the intent of this work or question we are seeking to address; instead we are looking to combine independent observations from a multi-instrument platform and explore the synergistic PBL information this approach can provide for boundary-layer height estimation, particularly via this fuzzy logic technique.

2. *The caption of Fig. 2 is still not clear. The caption should be able to roughly explain the figure so that readers do not need to read carefully in the text to understand the figure. For example, what are first- and second- generation variables? Those need to be clear in the caption. Fig. 2c and line 144, how is the temperature inversion height derived from the first-generation step as it only calculates the vertical wind variance? Fig. 2d, is it the inversion height or weighting function? 'inversion weight' is not defined in the text.*
To completely define what first and second generation steps are would be extensive for a figure caption that is only referring to the variables provided to the algorithm, regardless of the order of their use in the algorithm. We have never stated that inversion height is derived from the first generation step; it is used in the fuzzy logic algorithm as part of the first generation step. Please refer to the text in lines 139-150. In panel d, the figure shows the weight (value of the weighting function with respect to time) that the inversion height estimates gets in the fuzzy logic algorithm. To be more specific, some minor changes have been made to the figure label. Bolded text shows new additions.

*CLAMPS1 observations (collected 25 June 2020 in Norman, Oklahoma) used in the fuzzy logic algorithm. **Variables in** panels (a)–(d) are **used as** first generation variables, while **variables in** panels (e)–(j) are **used as** second generation variables. Each observed field is labeled with units **(if applicable) on its respective panel color bar.** Note that panel (a1) is a subset time window of panel (a) with a more narrow colorbar to highlight overnight vertical velocity variance values. Panel (b) is similar to panel (a) but considers only the high-frequency fluctuations in vertical velocity, as described in the text. Panels (c) and*

*(d) have local sunrise and sunset times marked for reference. Panel (d) is not an observed field, **but shows the magnitude of the** weighting **function** applied to inversion height, shown in panel (c), **which itself is computed from temperature profile data shown in (i)**.*

3. *(TR) Line 139-141: the logic is confusing here. It is indicated that a complete failure to detect BL height in the first-generation step occurs when the 'buoyant processes' dominates. If the first-generation step fails, the second-generation step is 'unable to recover'. However, according to the manuscript, the main advancement of the second-generation step is to improve 'buoyant processes' dominated (nighttime) BL height estimates. The question is that if the first-generation step failed and the second-generation is 'unable to recover', how does the second-generation step improve BL height estimates?*

As described in this part of the paper (and now also in our reply to the reviewer's comment 1), the first generation step in the overnight period includes inversion height, which is a thermodynamically derived property. The logic the reviewer describes as confusing is actually stating that it is important for us to include this capability and not only depend on mechanical processes overnight. Without including the inversion height, we would only allow the first generation estimate to be defined by mechanical processes detected by the Doppler lidar. At night in particular this is limiting. In the case no boundary-layer height was detected in the first-generation, the second-generation would have no basis from which to start (since it is required to work within a constrained distance of the first generation estimate). We know that the inversion height is really most important when mechanically-driven mixing processes are minimized/buoyant processes are dominant and this is the time the lidar detection methods are most likely to fail. We expect this to most often occur at night, so we assign a weighting function that only allows this first generation inversion height input to have an impact during the nighttime hours. During the second generation step, thermodynamic profiles are used at all times of the day, allowing buoyant processes (and other indicators that mixing has occurred, including well mixed wind profiles) to factor into boundary layer height estimation, within a constrained height range of the first generation estimate.

4. *The terminologies 'buoyant processes' and 'mechanical processes' are confusing. What exactly are 'buoyant processes' and what exactly are 'mechanical processes' in BL? Any references using these terminologies? In BL, we often use 'shear- or buoyant- driven' turbulence.*

In our view, the terms buoyant processes and mechanical processes are more encompassing. In other words, "processes" in this case would include (but not be limited to) turbulence. It would also include the effects of turbulence on atmospheric properties and structures that are measurable by our systems, which is the motivation behind our choice of language versus the perhaps more common "-driven" turbulence language often

found in theoretical, analytical, and numerical studies. We don't feel as comfortable using this terminology or feel it is quite fitting in our application as we are not truly measuring the turbulence, driven by either mechanical mixing or buoyancy (or its decay). We are able to detect properties or fields used as measures of turbulent processes like vertical velocity variance, which we use to infer the presence of turbulence or that turbulence is occurring, or to find indicators like well mixed layers, which we use to tell us turbulent mixing has likely occurred. To be consistent with what we can detect, we use the term processes -- which again could include turbulence and the larger scale mixing, warming, dying, cooling, moistening processes (which are more likely to be detectable by our sensors) that may occur as a result of turbulence. *To come closer to the terminology the reviewer prefers and finds more common, we have added the word 'driven' where applicable throughout the paper, but retained processes in most cases.*

5. *The manuscript states that 'the Harr wavelet transform is not sensitive to the dilation', which contradict with previous studies (e.g., Sawyer and Li, 2013). Why physically 'the Harr wavelet transform is not sensitive to the dilation' in this case? If it is not sensitive to the dilation length, why choose to use 100 m instead of, for example, 50 m or 300 m?*
   In a previous reply to the reviewer we discussed our findings regarding differences that we did find. There were some differences in the wavelet transform magnitude itself between dilation trials. However, in our application we subsequently apply search functions and conditions to the wavelet transform to identify the most prominent peaks. The outcomes of that search step—i.e., the identified peaks—produced results that were not sensitive to the dilation. Regarding the choice of 100m: as described in the manuscript dilation of 100m, 200m (as in B2018) and 300m were examined. Seeing no sensitivity to the outcome of the application, we saw no reason to increase (or decrease) the dilation size.

6. *The author's reply to my comment 6: 'The resulting vertical resolution varies with height, from O(10) m to O(100) m spacing; 300 m is the average.' It is not clear what does 'O(10) m' and 'O (100) m' mean. How come the average is 300 m?*
   The notation used in the paper is a style often called 'Big-O Notation,' where the O stands for 'Order of,'' meaning the whatever is being discussed can be expected to have magnitude of the order x, which is referenced in the O(x) usually with units; this can be a variety of scale types for a variety of use cases. Calling this style 'Big-O' notation is a naming convention most commonly referenced in data science and algorithm development fields, but it is seen in many applications outside of these communities. *We expected this notation style to be ubiquitous, but finding this may not be the case based on this reviewer's comment, we can simply extend the language to use the full 'order of' language.*

The dataset discussed here is the publicly available radiosonde data. To be clear, this is not a dataset we generated from our own observations. These data are collected and archived based on defined data requirements and structures as described in Schwartz and Govett (1992), which is referenced directly in the text. This reference and related summary in our text mentions that there were several reasons for the archive of data ending up with the format it has today when at one point in the past, multiple data sources were merged to create a unified archive. The average is 300 m because that is the average computed from this dataset which has variable vertical spacing, as described in the manuscript. In other words, if dz = the spacing between any two levels in a profile, and we know that dz varies throughout the profile, 300 is the average of all dz values in the profile. The minimum dz that occurs in the lowest levels of the atmosphere is on the order of 10s of meters. The maximum dz that occurs higher up in the atmosphere is on the order of 100s of meters.

7. *Line 315: From Fig. 6 and Fig. 7, it is not sufficient get the assertion that 'there is no indication from this analysis that using high-resolution data necessarily yields more accurate BL height values'. 1) Fig. 6 shows that the differences between high- and low resolution mainly occurs at small BL height values., e.g., during time. While Fig.7 shows the absolute BL height ranges for all cases. Small BL heights generally have smaller absolute BL height estimation ranges (across different methods), which large BL heights generally have large absolute BL height estimation ranges. Therefore, Fig. 7 is biased by the large BL heights. 2) there is a clear trend that High-res BL height estimations are much smaller than coarse-res BL height estimations for majority of the methods, as show in Fig.7. Therefore, it is hard to believe that 'coarse-resolution data' has neglected impacts on BL estimations.*

We find there is some potential confusion between the concepts referenced in this comment from the reviewer and what we include in our text and analysis. In this study, we have access to two sets of radiosonde data. One is higher resolution than the other, so for simplicity and clarity, that is the language we use throughout the paper. We do not intend to assign any specific connotation to a dataset by using these relative (high/coarse) terms. We must determine how to use the datasets, what data to use, etc. These choices are also important to consider in the context of comparing results from prior literature to our work (and vice versa in future works). How may the resolution of radiosonde data impact the analysis we do and what does that mean for the results and subsequent comparisons to other works? As we state in the paper, this is our motivation for these comparisons. As the reviewer points out, neither figure 6 or 7 in isolation is sufficient to provide all information relevant to the comparison between high and coarse resolution data. The reviewer states that 'it is not sufficient [to assert] that "there is no indication from this analysis that using high-resolution data necessarily yields more accurate BL height values"'. We disagree with this stance. We never state that there is no difference

between coarse and high-resolution sounding data or the BL depth estimates obtained from them; Figure 7 clearly shows otherwise for individual BL depth estimation methods. We do state that it is not obvious that the high-resolution estimates, being different from the estimates from coarse-resolution data, are intrinsically more correct, especially since small gradients can force detection events that appear unrelated to the BL-top. The manuscript also includes more information than what the reviewer referenced. This text is included below and the additional comments we make in the paper regarding this topic are bolded, including even a direct statement about our specific intention -- which nowhere states that coarse resolution data has neglected impacts on BL estimations. *Analyzing the methods separately highlights that using high-resolution sounding data tends to lead to lower median BL height, but the ranges and interquartile range values do not change much between coarse and high resolution datasets. There is no indication from this analysis that using high resolution data necessarily yields more accurate BL height values, and often users have access only to data that we have classified here as coarse resolution (i.e., publicly available and accessible National Weather Service radiosonde datasets). **Our intention with this analysis is to understand and consider possible implications of including different resolution sounding datasets. Moving forward, we use high resolution sounding data when it is available**. If it is not available a coarse resolution radiosonde profile is used instead.*

8. *For the answers to my comment #9: These conditions are also the major reason for many other BL estimates fail too. How does the first-generation BL estimates compare with sonde BL estimates if these conditions were excluded? The core question is still that: how does the author prove that including other data/measurements improves the BL estimation?*
Generally, we refer back to the first half of our response to comment 1 for a response to the question regarding the improvement of BL estimation derived from adding measurements. In the context of this comment, exclusion criteria are not linked to the first/second generation estimation steps. The exclusions were developed and applied based on conditions that were difficult for the instrumentation platforms considered to observe---including the reference radiosondes, thermodynamic instruments, *or* Doppler lidar. Any excluded condition would not represent the 'best case' comparison of a fully observed boundary-layer by a reference radiosonde and by *all* CLAMPS instruments, so it would be excluded from that set of comparisons.

9. *For the answers to my comment #10: 'deciding the threshold for 'what is a cloud' … is non-trivial'. Can't the DL backscatter intensity be used for cloud detection?*
Yes, the lidar can detect cloud bases, especially directly overhead. However, this would limit the definition of 'cloud' to optically-opaque clouds in the visible (technically near-infrared) portion of the EM spectrum. We know that cloud-like/near-cloud features

that are not quite opaque to the human eye are optically thick or even opaque in other portions of the EM spectrum and have important impacts on infrared and microwave radiation (thus potentially making impacts on the passive radiative sensor observations; we do include a 'cloud' flag from the lidar-based estimation in the thermodynamic retrieval, simply to aid the retrieval's optimal estimation in 'knowing' a cloud is there, but this would not be a robust criterion). With that knowledge, deciding when to or when not to classify periods as cloudy is non-trivial. It is also unclear how to handle mixed scenes when classifying cloud vs no-cloud in a binary way for this sort of exclusion task.

---

## Author Response (AR3)

**Response to reviewers for the paper** *A multi-instrument fuzzy-logic boundary-layer top detection algorithm* **by Elizabeth N Smith and Jacob T Carlin -** *Anonymous Reviewer # 2*

To guide the review process we have copied the reviewer comments in black, italicized text. Our responses are in regular blue font. We have responded to all the referee comments and, when appropriate to do so, shared the resulting alterations to our paper in a darker blue, italicized text.

*The authors' reply clarifies some of my comments. However, there still some confusing statements/definitions/discussions for me as listed below.*

*Comments:*

1. *While I concur that leveraging the synergy of multi-remote sensing boundary layer profiling measurements could enhance PBLH estimations by providing more constraining information, I find the authors' claim that 'the proposed method enables accurate boundary-layer height estimation both during daytime and nocturnal conditions' unconvincing based on the comparisons in Figures 8-11. The PBLHs estimated by remote-sensing (Fuzzy-logic) are consistently lower than those derived from radiosondes for PBLHs > 500 m or during afternoon convective boundary layer conditions (Figure 10). Why or what causes the underestimations from Fussy-logic PBLHs?*
   We make a statement that very closely matches the reviewer's observation lines 343-348. *From the bulk comparison shown in Fig 8, a few things become immediately apparent. The fuzzy logic method applied to CLAMPS observations is more likely to estimate a lower BL height than radiosondes. This is especially true in the afternoon hours, which correspond to 1800 and 2100 UTC for the area where these observations were collected. No 2100 UTC BL height comparisons (shown in yellow in Fig. 8) fall on or below the one-to-one line. The majority of 1800 UTC and most 0000 UTC BL height comparisons also trend above the one-to-one line, suggesting the fuzzy logic method applied to CLAMPS observations estimates a shallower BL. Earlier hours (0600 and 1200 UTC) do not demonstrate as much of a signal.* We then take a look at examples from CHEESEHEAD with more frequent soundings available to get a temporal comparison with radiosonde-based estimates in the following lines 348-355, confirming the prior statement. Then, to examine this further (i.e., answer the question posed in this comment) we do the exclusion analysis. This exclusion process was motivated specifically to better understand the potential causes of discrepancies between the radiosonde and CLAMPS-based BL height estimates. The exclusion criteria are described in the paper, and were used to successively exclude data pairs and get a better sense of algorithm performance. As each criteria was applied successively, pairs were removed from the comparison dataset and analysis was performed at each step, including statistics. This is all described in the paper in the final part of section 4. We can briefly review it here for more direct clarity and added commentary in the context of this review:

The first exclusion step eliminated cases that were deemed ambiguous without confident PBLH detection from radiosondes and/or when the detected PBLH appeared to be related to a different feature altogether (clouds, residual layers, etc). While this may appear 'subjective,' the dataset shown in 11a (after this exclusion) is similar to the dataset other studies might *start with*, never including such ambiguous cases (see, e.g., Banghoff et. al 2018-JTECH, Heinselman et al. 2009-JTECH). Removing these cases from the comparison did not present any pattern in the dataset, as expected.

As shown in Fig 11b, many of the pairs that indicated the fuzzy logic underestimated the PBLH relative to sondes were related to cases where CLAMPS observation platforms simply did not collect observations over a deep enough layer of the atmosphere (e.g., environments with relatively few available scatterers for the Doppler lidar, the deepest PBL cases resulting in important phenomena occurring in the upper, more coarse-resolution observing zone of the thermodynamic profiler, etc.). Users should be aware of the limitations of any instrument or dataset provided to any algorithm they use, and such limitations would likely apply to any remote-sensor, including lidar, based method. There are specialized lidar scanning strategies that could have been deployed (as in Bonin's work) and lidar post-processing algorithms, and special options in thermodynamic retrievals algorithms that could be applied by a user to attempt to increase the depth of the CLAMPS observed fields, but this is application specific and beyond the scope of understanding this algorithm.

Next, excluding cases where the radiosonde and CLAMPS-based methods identified different parts of the BL-top removed some more pairs mostly in the area just above the one-to-one line: instances when the fuzzy-logic method underestimated the PBLH relative to the sonde methods, but not by as large of a margin as in the previous set of excluded cases. Detecting different parts of PBLH can be connected to 1) ambiguity in defining what the PBLH is when the chosen defining parameters exist more as a *layer* (i.e., a relatively deep entrainment zone) than as a discrete level and 2) the methods by which different instrumentation senses the atmosphere and the PBLH estimation methods applied to those data. As discussed in the paper, the CLAMPS-based method is more likely to find a BL height closer to the surface since the fuzzy logic algorithm is a bottom-up algorithm that searches for the first point above the surface where the membership value crosses a threshold. Many points above this level could also be equivalent to this threshold value, but the algorithm will select the first one by design to make sure any PBLH detected is surface-based. The methods applied to radiosondes are, as discussed, variable. With regard to the instrumentation, thermodynamic retrievals will not detect shallow, minor gradients that radiosondes might. Similarly, sondes may also be more apt than the retrieval to capture thermodynamic structures associated with deep PBL top zones (e.g., double inversions, shallow elevated isothermal layers, etc.). Accordingly, CLAMPS is most likely to capture the *bottom* of these zones, while various methods applied to sondes may result in different representations.

The final exclusions are cases with complex structures in the PBL as determined by examining the radiosonde profiles and CLAMPS observations---not the PBLH estimates. As described in the case of detecting different parts of the PBL top, CLAMPS thermodynamic retrievals are not likely to fully represent shallow gradients due to the optimal estimation approach. In the event the complex structure includes saturated layers near the surface, the thermodynamic profiler data may not include as much independent information for the retrieval process, adding even more uncertainty. This is well known and documented in the literature concerning the retrieval and in dataset documentation. Additionally, the various PBLH techniques applied to radiosondes are not necessarily well suited for profiles with complex temperature and moisture gradients. For these reasons, we excluded these pairs in the final round of exclusions---seeing no clear pattern in the removed data which was expected---leaving a dataset that we found most representative of cases where the observing conditions were generally ideal for examining the behavior of the *algorithm* without conflating it with variability of observation quality across cases. That dataset is shown in panel 11d. While many, if not most, of the instances of overestimation have been excluded, some cases where the fuzzy logic algorithm applied to CLAMPS observations leads to an underestimation of BL height relative to sondes persist. There is not an immediate clear pattern to these pairs. The majority of the pairs are in the later half of the diurnal cycle, but they span all three considered locations. An obvious reason for this persistence is not yet clear. As stated in the paper, more frequent and regularly available radiosondes are required to extend this analysis in order to deepen understanding about the impact time of day has on the performance of the fuzzy logic algorithm itself. It would also be interesting to add more locations with profiler-sonde comparisons to evaluate the impacts of moisture in these comparisons.

2. *Based on Figure 2, it appears that the second-generation step yields even lower PBLHs than the first-generation step. This suggests that the second-generation step further underestimates PBLHs under conditions of a convective boundary layer. Is that correct?* As stated in the prior response, on any one day at any one observation time, we could choose to show a 'better' or 'worse' comparison to either generation estimate. Forming arguments or opinions about data from a single comparison point is not a particularly informative method and not the method we choose to determine the 'performance' of any technique. Comparing with the estimates from the first and second generation is not the intent of this work or question we are seeking to address; instead we are looking to combine independent observations from a multi-instrument platform and explore the synergistic PBL information this approach can provide for boundary-layer height estimation, particularly via this fuzzy logic technique. Further, decreasing the height of the PBLH estimate is not necessarily a degradation---there are scenarios in which lidar-based estimates could be too high (e.g., elevated cloud bases). Setting that aside for now and following the reviewer down this path for a moment, taking a closer at

20200625 (Fig 3) we can see that the 2nd generation estimate does not always produce lower estimates. See the period between 16 and approximately 18 UTC, from 2030 to 21 UTC and several of the individual estimates without the ten-min triangle smoothing applied (dots), which are sometimes deeper estimates than their first-generation counterparts. Similar behavior is apparent on 20190926 (Fig 10), where the period from 15 to 19 UTC and from 10-11:30 UTC. At all times of the day, the second generation step adds synergistic observations. This can include more information from the lidar and information beyond what a lidar can provide. This manuscript fundamentally intends to introduce an algorithm with this capability. If users are interested in lidar-only methods, those are already available---eliminating the second generation would effectively turn this algorithm into a repackage of those techniques.

3. *The comparisons between Fuzzy-logic and radiosonde PBLHs do not seem as robust as those shown in previous studies that exclusively used Doppler lidar measurements, such as those in Tucker et al. (2009) and Krishnamurthy et al. (2021). Could you provide the correlation coefficients between Fuzzy-logic PBLHs and Radiosonde PBLHs as depicted in Figures 8, 9, and 11?*
*References:*
*Tucker, S. C., C. J. Senff, A. M. Weickmann, W. A. Brewer, R. M. Banta, S. P. Sandberg, D. C. Law, and R. M. Hardesty, 2009: Doppler Lidar Estimation of Mixing Height Using Turbulence, Shear, and Aerosol Profiles. J. Atmos. Oceanic Technol., 26, 673–688, https://doi.org/10.1175/2008JTECHA1157.1.*
*Krishnamurthy, R., Newsom, R. K., Berg, L. K., Xiao, H., Ma, P.-L., and Turner, D. D.: On the estimation of boundary layer heights: a machine learning approach, Atmos. Meas. Tech., 14, 4403–4424, https://doi.org/10.5194/amt-14-4403-2021*

The requested analysis was part of the original manuscript and has never been removed. Please see table 3, values reported in Fig. 9 and discussion of values throughout section 4. Re: Tucker et al 2009, we would expect those comparisons to match more closely than ours for a couple of reasons. 1) the lidar system used (HRDL) operates at a different wavelength and generally a bit differently than the system we used, so it often collects stronger returns from aerosol backscatter and can profile slightly deeper into the atmosphere 2) the approach described defaults to defining the mixing layer height using empirically derived thresholds of vertical velocity variance, but in the event an estimate cannot be made this way at a given time, the next best option is selected based on the environmental conditions, then finally a manual check to 'eliminate problematic data and/or unexplained discontinuities.' Re: Krishnamurthy et al. 2021: This is a random forest approach using a large number of inputs that is trained and tested at one location (without considering missing data, which is a common problem and focus of research in machine learning). If you examine their Figure 3, you will see some different perspectives on Tucker-based comparisons. Even with the largely different approaches and the more varied regimes of the data we include, the correlations we report from the

exclusion analysis in our Table 3 are generally comparable to the correlations shown in their Figure 5.

4. *The authors have noted that in the first-generation step, the temperature inversion height was used as an input. Could you clarify how the temperature inversion height is derived? What degree of temperature difference is considered to constitute an inversion? The term 'temperature inversion height' is introduced in line 139 without an explanation of how it is obtained. In the response, the authors claim, 'We have never stated that inversion height is derived from the first-generation step.' If so, could you specify when the 'inversion height' is derived?*

   The temperature inversion has been included in the algorithm and in the description of the algorithm in all versions of the manuscript. CLAMPS provides temperature profiles in the form of thermodynamic retrievals from infrared spectrometers and/or microwave radiometers (depending on instrument availability). The retrieval using an optimal estimation approach, which is an ill posed problem. This algorithm and its outcomes have been well documented and published over many years. We know that retrieved profiles are unlikely to represent shallow or minor gradients. As such, we compute the gradient directly as dT/dZ and find the inversion height as the first level where dT/dz reverses sign (e.g., becomes positive) above the surface and below 1.5 km, since elevated inversions are of interest for this application. Since the algorithm output is not expected to depict minor fluctuations in the thermodynamic profiles, there is no required 'threshold' for dT/dz to qualify as an inversion.

5. *Typically, a figure caption should provide a brief explanation of what the figure represents, such as the observations or variables depicted in Figure 2. This would provide readers with basic information, and I don't believe it would be as 'extensive' as the authors suggested in their response. They could relocate the detailed discussions or illustrations of the figure to the main body of the text.*

   The caption has been revised again. *Time-height cross-sections showing the diurnal evolution of CLAMPS1 observations collected on 25 June 2020 in Norman, Oklahoma which are used in the fuzzy logic algorithm to produce the aggregate fields and BL height estimates shown in Figure 3. Panels (a-c) show the variables used in the first-generation step of the fuzzy-logic algorithm [(a) vertical velocity variance, (b) high-frequency vertical velocity variance, (c) temperature inversion height], with the weighting given to panel (c) shown in (d) based on sunrise/sunset time, while panels (e-j) show the variables used in the second-generation step of the fuzzy-logic algorithm [(e) backscatter intensity, (f) backscatter intensity variance, (g) u-wind, (h) v-wind, (i) temperature, and (j) water vapor mixing ratio]. Panel (a) includes a subsetted window (a1) between 00—12 UTC with a finer colorscale to show overnight values of vertical velocity variance. Each observed field is labeled with units (if applicable) on its respective panel color bar.*

6. *Despite several rounds of discussion, the precise definitions of 'buoyancy-driven processes' and 'mechanically-driven processes' remain unclear*

The Tucker et al. 2009 paper the reviewer referenced uses yet another version of terminology: "shear-driven boundary-layer." At this stage, after many rounds of discussion, the authors respectfully propose we agree to disagree in the absence of obvious community defined norms.

7. *The authors replied that 'if dz = the spacing between any two levels in a profile, and we know that dz varies throughout the profile, 300 is the average of all dz values in the profile. The minimum dz that occurs in the lowest levels of the atmosphere is on the order of 10s of meters. The maximum dz that occurs higher up in the atmosphere is on the order of 100s of meters.' If a group of numbers have a minimum of 10 and maximum of 100, how can the average to be 300?*

We clearly state **on the order of 100s of meters**, not that the maximum is 100 m. The "order" of 100 meters can include values as low as 100 m and up to and including 999 m. Written in mathematical interval notation, this would be expressed as [100,999]. The average value of dz can therefore be 300 m since 100 m < 300 m < 999 m.

8. *As for the comparison of high- vs. low-resolution sounding data, the authors state, 'Analyzing the methods separately highlights that using high-resolution sounding data tends to lead to lower median BL height, but the ranges and interquartile range values do not change much between coarse and high resolution datasets. There is no indication from this analysis that using high resolution data necessarily yields more accurate BL height values'. It appears that the authors are suggesting that because 'the coarse and high resolution datasets have similar PBLH ranges and interquartile range values', therefore, 'There is no indication from this analysis that using high resolution data necessarily yields more accurate BL height values'. I don't believe this is a solid conclusion. For instance, under an extreme condition where the coarse sounding data produce identical PBLHs, resulting in ranges and interquartile range values of zero, it certainly doesn't imply that coarse sounding data provides accurate PBLH estimations.*

The reviewer provides a hypothetical condition "where the coarse sounding data produce identical PBLHs, resulting in ranges and interquartile range values of zero." We are interpreting this to mean the coarse data lead all applied methods for estimating PBLH to output the same result. It is unclear why this hypothetical "extreme condition" would only impact the coarse resolution data. Further, we state directly that "there is no indication *from this analysis*" meaning in the data presented in the paper. Those data do not include such a condition. Most fundamentally, we aren't sure exactly how this would be connected to our assertion that, based on our analysis, we cannot make any clear statement about one dataset being more accurate than the other. That assertion has less to do with the IQR of each method and is more about the fact that, absent any other independent source of data besides the radiosondes, it is not clear that the pros/cons of both the hi-res and coarse soundings (e.g., methods applied to the coarse data may not be misled by small fluctuations, but important small details may similarly be missed in other cases) necessarily mean one is better than the other overall. In any case, to hopefully

provide the assurance that is perhaps needed to resolve this concern we have performed additional testing on these comparisons. We used bootstrap hypothesis testing to estimate the confidence interval of test statistics by repeatedly calculating the test statistics on bootstrapped samples. If the test statistic falls outside of the resulting confidence interval, then we reject our hypothesis. Our hypothesis in this case is that the coarse-resolution and high-resolution datasets are similar for any given radiosonde-based PBLH estimate method. For each method, we performed a 10,000 bootstrap resampling analysis twice to compare the median and IQR as the test statistic for which we defined confidence intervals with 95% confidence. In all instances (that is for all methods in terms of the median and the IQR) the test statistic fell within the confidence interval, indicating that, regardless of the technique, differences between PBLH estimates based on coarse- and high-resolution are not statistically significant. The following was added to the text:

*Analyzing the methods separately highlights that using high-resolution sounding data tends to lead to lower median BL height, but the ranges and interquartile range values do not change much between coarse and high resolution datasets. To be certain, we conducted 10,000 bootstrap resampling analyses for each method to evaluate if the median and interquartile range values were significantly different between high- and coarse-resolution datasets. We found with 95% confidence that there were no statistically significant differences. There is no indication from this analysis that using high-resolution data necessarily yields more accurate BL height values, and often users have access only to data that we have classified here as coarse resolution (i.e., publicly available and accessible National Weather Service radiosonde datasets). Our intention with this analysis is to understand and consider possible implications of including different resolution sounding datasets. Moving forward, we use high resolution sounding data when it is available. If it is not available a coarse resolution radiosonde profile is used instead.*